# Bridging Causality, Individual Fairness, and Adversarial Robustness in the Absence of Structural Causal Model

**Ahmad-Reza Ehyaei**                                                   *ahmad.ehyaei@tuebingen.mpg.de*
*Max Planck Institute for Intelligent Systems Tübingen AI Center,*
*Tübingen, Germany*

**Golnoosh Farnadi**                                                            *farnadig@mila.quebec*
*Mila - Québec AI Institute, Université de Montréal McGill University,*
*Montréal, Canada*

**Samira Samadi**                                                       *ssamadi@tuebingen.mpg.de*
*Max Planck Institute for Intelligent Systems Tübingen AI Center,*
*Tübingen, Germany*

**Reviewed on OpenReview:**   *https://mailhost.tuebingen.mpg.de/SRedirect/B8BAABAE/openreview.net/forum?id=2nRcWy3RLM*

## Abstract

Despite the essential need for comprehensive considerations in responsible AI, factors such as robustness, fairness, and causality are often studied in isolation. Adversarial perturbation, used to identify vulnerabilities in models, and individual fairness, aiming for equitable treatment of similar individuals, despite initial differences, both depend on metrics to generate comparable input data instances. Previous attempts to define such joint metrics often lack general assumptions about data and were unable to reflect counterfactual proximity. To address this, our paper introduces a *causal fair metric* formulated based on causal structures encompassing sensitive attributes and protected causal perturbation. To enhance the applicability of our metric, we introduce metric learning as a method for estimating and deploying the metric in real-world scenarios, even when structural causal models are unavailable or when causal structures cannot be identified. We also demonstrate the applications of the causal fair metric in classifiers. Empirical evaluation of real-world and synthetic datasets illustrates the effectiveness of our proposed metric in achieving an accurate classifier with fairness, resilience to adversarial perturbations, and a nuanced understanding of causal relationships.

## 1 Introduction

While fairness, robustness, and causality are central to responsible AI, they are often studied in isolation despite the need for systems to address them all comprehensively. In this work, however, we demonstrate that individual fairness and adversarial robustness, both of which rely on metrics to produce comparable data instances, are interconnected and can be learned simultaneously. On one hand, the concept of *individual fairness*, as defined by Dwork et al., 2012 focuses on the fair treatment of similar individuals to prevent discrimination based on individual characteristics. The definition of individual fairness, whether through the Lipschitz formulation (Dwork et al., 2012) or the $\epsilon - \delta$ method (John et al., 2020), requires the creation and assessment of a *fair metric*. These metrics are essential quantitative tools for evaluating whether algorithms adhere to the principles of individual fairness.

On the other hand, *Adversarial perturbation*, as outlined by (Goodfellow et al., 2014) and (Madry et al., 2017), involves the purposeful manipulation of input data to uncover machine learning model vulnerabilities or assess robustness. This concept is closely related to *metrics* that measure the impact of changes in input

data on model performance. Often, it involves using distance metrics to quantify the differences between original and perturbed inputs.

In this work, we show that metrics for both individual fairness and adversarial robustness can be simultaneously defined through the lens of causality to better reflects the true characteristics of the underlying data. When dealing with a causal structure underlying data, traditional metrics like the Euclidean norm, fail to account for causal relationships, as noted by Kilbertus et al. (Kilbertus et al., 2017). This limitation becomes especially evident when aiming for fair treatment for sensitive attributes. In such scenarios, the most suitable metric would be one that generates minimal values for counterfactual instances associated with each data point.

Previous research frequently simplified counterfactual calculations by only modifying levels of sensitive features. In the work by Dominguez et al. (Dominguez-Olmedo et al., 2022), adversarial perturbations were integrated into structural causal models (SCM), with a primary focus on continuous features, which may have neglected aspects of fairness. On the other hand, Ehyaei et al. (Ehyaei et al., 2023b) developed a fair metric based on functional causal structure, tailored to safeguard sensitive attributes. However, their approach is limited to specific examples and lacks support from a well-established theoretical foundation.

In this study, we aim to bridge the gap between fair metrics in causal structures and sensitive attributes. We begin by identifying suitable properties that align with our objectives and subsequently derive this metric from observational data without knowing the causal structure assumption. In Section 4, we first introduce a definition for a *causal fair metric*, effectively addressing both causality and the protection of sensitive attributes. Next, in Section 5, we use the causal fair metric to create a *protected causal perturbation*, enhancing adversarial perturbation with causality and sensitivity considerations. We also examine its geometric properties and attributes. Constructing a causal fair metric typically requires knowledge of SCM, which is often unavailable in many real-world applications. To overcome this limitation, we propose to derive the metric from data. In Section 6, we illustrate that relying solely on observational or interventional data is insufficient for learning the causal fair metric. To address the absence of SCMs, an alternative approach involves metric learning using tagged distance data. These tags indicate proximity values or labels indicating data point closeness. By discussing the requirements of other methods, we focus on deep metric learning due to its compatibility with the structure of causal fair metrics. To enhance practicality, we employ contrastive and triplet deep metric learning scenarios. This approach demonstrates that the causal fair metric is learnable even in the absence of a known causal structure or when causal detection is impossible in certain models. Finally, in Section 8, through experiments on both synthetic and real-world datasets, we empirically verify our theoretical findings. Our results illustrate that knowing the structure of the causal fair metric amplifies learning performance within deep metric learning scenarios. Furthermore, our empirical analysis reveals that label-based metric approaches strike a practical balance between applicability and accuracy and are more aligned with the concept of protected causal perturbation. To demonstrate the effectiveness of our framework, we incorporate our empirical causal fair metric into a fairness learning method for classifiers. Unlike existing approaches that require knowledge of the causal structure, ours operates without SCMs, yet yields notable enhancements in fairness while preserving accuracy. In summary, our main contributions are:

- **Causal Fair Metric (§4):** We present a causal fair metric that incorporates both causal considerations and the protection of sensitive attributes. In addition, we demonstrate how our proposed causal fair metric can be embedded in exogenous space.

- **Protected Causal Perturbation (§5):** We use our proposed causal fair metric to generate adversarial perturbation within causal structures while addressing fairness concerns.

- **Causal Fair Metric Learning (§6):** Theoretically, we show that, without SCM assumptions, learning a fair metric from observational or interventional distributions is not guaranteed. We introduce a learning algorithm designed to extract a causal fair metric from empirical data, with a focus on both causality and fairness considerations.

- **Fairness, Robustness and Causality Classifier (§ 7):** To demonstrate the effectiveness of our approach, we apply it to a classification problem and introduce **ECAPIFY**, which combines metric learning and fair learning without relying on a known SCM.

## 2 Related Work

In this section, we explore previous research defining metric learning, whether for adversarial perturbation or individual fairness. The most relevant study to ours is conducted by Ehyaei et al. (Ehyaei et al., 2023b), which worked on constructing a fair metric in the presence of causal structures and sensitive attributes. However, unlike ours, their metric is limited to a specific family of dissimilarity functions and lacks a comprehensive characterization of its properties. In Mukherjee (Mukherjee et al., 2020), the authors attempted fair metric learning, but their method didn't heavily rely on causal structure. They assumed an embedding into a space where sensitive attributes form a linear subspace but didn't clarify its connection to SCM. Moreover, they assumed knowledge of the embedding map during metric learning. In Ilvento (Ilvento, 2020), submetrics were developed for learning metrics for individual fairness using human judgments. Under specific assumptions about point distribution and representative point selection, these submetrics maintained accuracy relative to the true metric. However, this work didn't address the impact of sensitive attributes, which often compromise metric properties. Spectral-based metric estimation methods, akin to those in Zhang (Zhang et al., 2016) and Olson (Olson, 2022), often require specific embedding kernel forms or observations of all pairwise distances $d(v_i, v_j)$ for guaranteed metric convergence. Fair representation learning (Zemel et al., 2013; McNamara et al., 2017; Ruoss et al., 2020) aims to map indiviuals to prototypes. Their primary aim frequently involves eliminating protected attributes while preserving performance-relevant information during the training phase. Another non-linear metric estimator is the tree-based approach, proposed by Demirovic (Demirović & Stuckey, 2021). They introduced a novel algorithm using bi-objective optimization to compute decision trees that are provably optimal for non-linear metrics. Online learning algorithms, as in (Bechavod et al., 2020; Gillen et al., 2018), ensure a finite number of fairness constraint violations and bounded regret, relying on some metric-based assumptions. Various spectral, probabilistic, and deep metric learning methods are discussed in (Ghojogh et al., 2022; 2023; Suárez et al., 2021; Francis & Raimond, 2021). To the best of our knowledge, none of the existing algorithms address the integration of causal structure and sensitive attributes in metric learning.

## 3 Background

**Structural Causal Model.** A SCM for a set of $n$ random variables $\mathbf{V} = \{\mathbf{V}_i\}_{i=1}^n$ is represented by the tuple $\mathcal{M} = \langle \mathcal{G}, \mathbf{V}, \mathbf{U}, \mathbb{F}, \mathbf{P}_{\mathcal{U}} \rangle$ (Pearl, 2009), where:

- The set $\mathbb{F} = \{\mathbf{V}_i := f_i(\mathbf{V}_{\text{Pa}(i)}, \mathbf{U}_i)\}_{i=1}^n$ contains *structural equations*, with each equation $f_i$ denoting the causal connection between the endogenous variable $\mathbf{V}_i$, its direct causal parents $\mathbf{V}_{\text{Pa}(i)}$ from $\mathcal{G}$, and an exogenous variable $\mathbf{U} = \{\mathbf{U}_i\}_{i=1}^n$ signifying unobservable background influences. In this work, we suppose $\mathcal{G}$ is a directed acyclic graph.
- The distribution $\mathbb{P}_{\mathbf{U}}$ of exogenous noise variables factorizes, $\mathbb{P}_{\mathbf{U}} = \prod_{i=1}^n \mathbb{P}_{\mathbf{U}_i}$, due to the assumption of *causal sufficiency*.

Under acyclicity, each instance $u \in \mathcal{U}$ of the exogenous space $\mathcal{U}$ uniquely determined by $v \in \mathcal{V}$ with the reduced-form mapping $g : \mathcal{U} \to \mathcal{V}$, where $g$ is obtained by iteratively substituting the structural equations $\mathbb{F}$ following the causal graph's topological order $\mathcal{G}$. The SCM entails a unique joint distribution $\mathbb{P}_X$ over the endogenous variables through the reduced-form mapping, $\mathbb{P}_{\mathbf{V}}(\mathbf{V} = v) := \mathbb{P}_{\mathbf{U}}(\mathbf{U} = g^{-1}(v))$ where $g^{-1}$ is the preimage of $g$.

**Causal Identifiability.** Discovering true causal connections among variables solely from observational data typically necessitates additional assumptions about the structural functions $\mathbb{F}$. One identifiable family of SCMs is the additive noise model (ANM) (Hoyer et al., 2009), represented by $\mathbf{V} = f(\mathbf{V}) + \mathbf{U}$. In ANMs, obtaining the relationship from $u$ to $v$ is straightforward when considering $I$ as the identity function ($I(v) = v$), then $g$ is obtained by $g = (I - f)^{-1}$. Post-nonlinear models (Zhang & Hyvarinen, 2012) and location-scale noise models (Immer et al., 2023) are other identifiable SCM families.

**Interventions.** SCMs facilitate modeling and assessing the impact of external manipulation on the system represented by the intervention (Peters et al., 2017). Two main intervention types are *hard interventions*

and *soft interventions*. In hard interventions (expressed as $\mathcal{M}^{do(\mathbf{V}_{\mathcal{I}}:=\theta)}$), a subset $\mathcal{I} \subseteq \{1, \ldots, n\}$ of features $\mathbf{V}_{\mathcal{I}}$ is forcibly fixed to a constant $\theta \in \mathbb{R}^{|\mathcal{I}|}$ by excluding relevant parts of the structural equations:

$$\mathbb{F}^{do(\mathbf{V}_{\mathcal{I}}:=\theta)} = \begin{cases} \mathbf{V}_i := \theta_i & \text{if } i \in \mathcal{I} \\ \mathbf{V}_i := f_i(\mathbf{V}_{\mathbf{Pa}(i)}, \mathbf{U}_i) & \text{otherwise} \end{cases}$$

Hard interventions disrupt the causal connections between affected variables and their ancestral components in the causal graph, whereas soft interventions maintain all causal relationships while adjusting the structural equation functions. For example, *additive (shift) intervention* (Eberhardt & Scheines, 2007), denoted as $\mathcal{M}^{do(\mathbf{V}_{\mathcal{I}}+=\delta)}$, modify features $\mathbf{V}$ using a perturbation vector $\delta \in \mathbb{R}^n$ with equations $\left\{ V_i := f_i\left(\mathbf{V}_{\mathbf{Pa}(i)}, \mathbf{U}_i\right) + \delta_i \right\}_{i=1}^n$.

**Counterfactuals.** *Counterfactual* is a hypothetical scenario that represents what would have happened if certain interventions or changes were applied to the variables in the SCM. The counterfactual outcome $\mathbf{CF}(v, \theta)$ for a specific variable $\mathbf{V}_{\mathcal{I}}$ under the hard intervention $do(\mathbf{V}_{\mathcal{I}} := \theta)$ can be computed using the modified structural equations as $g^\theta(g^{-1}(v))$, where $g^\theta$ represents the altered reduced-form mapping $\mathcal{M}^{do(\mathbf{V}_{\mathcal{I}}:=\theta)}$ after the intervention.

**Sensitive Attribute.** A sensitive attribute, like race, holds ethical or legal significance in decision-making, such as in hiring, lending, or criminal justice, determining equitable treatment or outcomes for individuals or groups. Let $\mathbf{S} \in \{\mathbf{V}_1, \ldots, \mathbf{V}_n\}$ represent a sensitive attribute with domain $\mathcal{S}$ (discrete or continuous). For each instance $v \in \mathcal{V}$, the set of *counterfactual twins* regarding the sensitive feature $\mathbf{S}$ is obtained by $\ddot{\mathbf{v}} = \{\ddot{v}_s = \mathbf{CF}(v, s) : s \in \mathcal{S}\}$.

**Individual Fairness.** *Individual fairness*, as introduced by Dwork et al., 2012, ensures equitable treatment for individuals with comparable predefined metric similarities. Two formulations, including the Lipschitz mapping-based formulation (Dwork et al., 2012):

$$d_{\mathcal{Y}}(h(v), h(v')) \leq L \, d_{\mathcal{V}}(v, v') \quad \forall v, v' \in \mathcal{V}$$

and the $\epsilon$-$\delta$ formulation (John et al., 2020):

$$\forall v, v' \in \mathcal{V} \quad d_{\mathcal{V}}(v, v') \leq \delta \implies d_{\mathcal{Y}}(h(v), h(v')) \leq \epsilon$$

have been proposed. Where, $d_{\mathcal{X}}$ and $d_{\mathcal{Y}}$ are metrics for the input and output spaces, respectively, with $h$ as the classifier and $L \in \mathbb{R}_+$. The essence of the definition is centered around the *fair metric $d_{\mathcal{X}}$*, which measures individual similarity based on relevant attributes.

*Counterfactual fairness*, as introduced by Kusner et al. (Kusner et al., 2017), defines fairness using causal models. This approach compares an individual's actual outcomes with hypothetical outcomes in a scenario where sensitive features differ. A classifier $h$ is deemed counterfactually fair if it satisfies the following condition, $h(\ddot{v}_s) = h(\ddot{v}_{s'}) \quad \forall s, s' \in \mathcal{S}$.

## 4  Fair Metric

The fair metric, often used in previous studies, becomes ambiguous when applied to problems involving causal structures and sensitive attributes (Ghojogh et al., 2022). To clarify the necessary properties of a fair metric in these contexts, consider the following example.

**Example 4.1** *Consider two SCMs, $\mathcal{M}$ and $\mathcal{M}'$, describing gender ($\mathbf{G}$), income ($\mathbf{I}$), and education ($\mathbf{E}$). $\mathcal{M}$ models these variables as independent, while $\mathcal{M}'$ specifies a linear causal relationship:*

$$\mathcal{M} = \begin{cases} \mathbf{G} := \mathbf{U}_G, \\ \mathbf{E} := \mathbf{U}_E, \\ \mathbf{I} := \mathbf{U}_I \end{cases}, \quad \mathcal{M}' = \begin{cases} \mathbf{G} := \mathbf{U}_G, \\ \mathbf{E} := \mathbf{G} + \mathbf{U}_E, \\ \mathbf{I} := \mathbf{G} + 2\mathbf{E} + \mathbf{U}_I \end{cases}, \quad \mathcal{U} = \begin{cases} \mathbf{U}_G \sim \mathcal{B}(0.5) \\ \mathbf{U}_E \sim \mathcal{N}(0, 1) \\ \mathbf{U}_I \sim \mathcal{N}(0, 1) \end{cases}$$

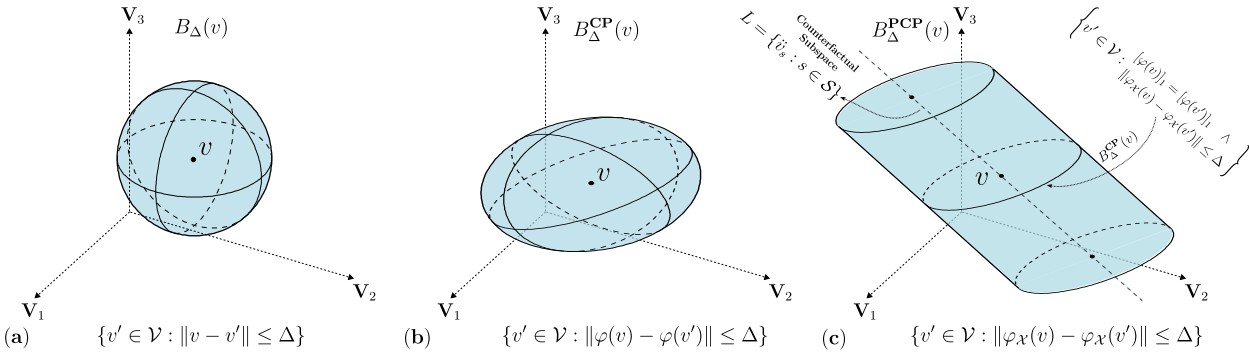

Figure 1: illustrates the progression from a basic perturbation to a protected causal perturbation ball. Consider a simple linear SCM with Euclidean norm in both exogenous and endogenous spaces. (a) a perturbation ball that does not account for causality or the protection of sensitive features, (b) a perturbation ball that includes causality but assumes the absence of sensitive features, and (c) The counterfactual perturbation space, created using a counterfactual space based on a sensitive attribute, can be visualized as the axis $L$ of a cylinder. Surrounding ellipses represent a causal perturbation ball $B_\Delta^{\mathbf{CP}}$, encompassing perturbations of non-sensitive with radii $\Delta$.

*Here, $\mathbf{U}_G$ represents the gender distribution, while $\mathbf{U}_E$ and $\mathbf{U}_I$ are intrinsic talents for education and income, respectively. Consider the $d(\mathbf{V}, \mathbf{V'}) = |\mathbf{E} - \mathbf{E'}| + |\mathbf{I} - \mathbf{I'}|$ $L_1$-norm on non-sensitive attributes to compare individuals. If two individuals have less than a 0.2 unit difference, they are deemed similar. For an individual with data $v = (M, 1, 2)$ (M = 1 for Male), a perturbation in education by 0.1 units ($\Delta = (0, .1, 0)$) in $\mathcal{M}$ results in $\mathbf{CF}(v, \Delta) = (M, 1.1, 2)$, which is similar to $v$. In $\mathcal{M'}$, $\mathbf{CF}(v, \Delta) = (M, 1.1, 2.2)$ gives a distance of 0.3, indicating dissimilarity. To protect against gender bias, individuals with the same intrinsic characteristics but different genders should behave similarly. This is modeled by a counterfactual change in gender. In $\mathcal{M}$, $\mathbf{CF}(v, F) = (F, 1, 2)$, so $v$ and its twin are similar because $d(v, \ddot{v}_F) = 0$. However, in $\mathcal{M'}$, $\mathbf{CF}(v, F) = (F, 0, -1)$, resulting in $d(v, \ddot{v}_F) = 4$, which indicates dissimilarity.*

The above example shows that combining a causal structure with sensitive attributes requires a dissimilarity function that remains zero for counterfactual twins and is stable against small changes in non-sensitive attributes. Creating counterfactual twins is straightforward, but defining small changes for non-sensitive features needs further exploration. Following Dominguez-Olmedo et al., 2022 and Ehyaei et al., 2023a, additive interventions are used as perturbations in additive noise models. Now we are ready for a proper definition of a causal fair metric.

**Definition 4.2 (Causal Fair Metric)** *Let $d : \mathcal{V} \times \mathcal{V} \to \mathbb{R}_{\geq 0}$ represent a metric defined on the feature space $\mathcal{V}$, generated by a SCM $\mathcal{M}$. Let $\mathbf{S}$ denote a sensitive attribute, and $\mathcal{I}$ represent the index set of sensitive features within the SCM. The metric is called a causal fair metric if it adheres to the following properties:*

*(i) For all $v \in \mathcal{V}$ and $s \in \mathcal{S}$, the metric is zero only for twin pairs, i.e., $d(v, \ddot{v}_s) = 0$.*

*(ii) For every $v \in \mathcal{V}$ and any $\delta > 0$, there exists $\epsilon$ such that for any sufficiently small intervention ($\|\Delta\| \leq \epsilon$) on the non-sensitive attributes, the distance $d(v, \mathbf{CF}(v, \Delta))$ remains less than $\delta$.*

The first property highlights that a fair metric maintains counterfactual fairness, meaning the distance between an instance and its counterfactual is zero. The second property ensures that the intuition of similarity in the exogenous space is inherited by the feature space, allowing us to set thresholds to define similarity effectively.

Constructing a metric on $\mathcal{M}$ is challenging because it must account for causal relationships in its dissimilarity function. By applying the causal sufficiency principle, which ensures feature independence in the exogenous space, we can define individual similarity functions for each feature's noise. These individual metrics enable us to construct a holistic metric in the noise space, which is then extended to the feature space through a push-forward metric, i.e., $d_{\mathcal{V}}(v, v') = d_{\mathcal{U}}(g^{-1}(v), g^{-1}(v'))$. We assume $g$ is invertible, as required for identifiability and counterfactual identifiability. Bijective generation mechanisms (BGMs), including additive noise models, satisfy these conditions. However, this approach is inadequate because $d$ must be defined for

every counterfactual of $v$. Generally, we have $\text{Range}(\mathcal{M}) \subset \bigcup_{s \in \mathcal{S}} \text{Range}(\mathcal{M}^{do(\mathbf{A} := s)})$. Therefore, we need to consider a space that encompasses all counterfactual values.

**Definition 4.3 (Semi-latent Space (Ehyaei et al., 2023b))** *Consider $\mathcal{M}$ with sensitive attributes indexed by $\mathcal{I}$. The converted SCM denoted as $\mathcal{M}^{\mathcal{S}}$, is derived from $\mathcal{M}$ by removing the causal effects of parents of sensitive attributes and replacing their exogenous variables with endogenous ones. The structural equations for $\mathcal{M}^{\mathcal{S}}$ are as follows:*

$$\mathbf{V}_i^{\mathcal{S}} := \begin{cases} \mathbf{V}_i & i \in \mathcal{I} \\ f_i(\mathbf{V}_{\boldsymbol{pa}(i)}) + \mathbf{U}_i & i \notin \mathcal{I} \end{cases}$$

*The exogenous space corresponding to $\mathcal{M}^{\mathcal{S}}$, denoted by $\mathcal{Q}$, includes the sensitive attributes and the non-sensitive parts of the exogenous variables of $\mathcal{M}$. This space called the semi-latent space, is constructed as $\mathcal{Q} = \mathcal{S} \times \mathcal{U}_{\mathcal{X}}$, where $\mathcal{U}_{\mathcal{X}}$ is the non-sensitive part of the exogenous space. There map bijective map $\varphi : \mathcal{V} \to \mathcal{Q}$ from feature space to the semi-latent space by the below formulation:*

$$\varphi_i(v) := \begin{cases} v_i & i \in \mathcal{I} \\ (g^{-1})_i(v) & i \notin \mathcal{I} \end{cases}, \quad \varphi_i^{-1}(u) := \begin{cases} u_i & i \in \mathcal{I} \\ f_i(\varphi_{\boldsymbol{pa}(i)}^{-1}(u)) + u_i & i \notin \mathcal{I} \end{cases} \tag{1}$$

where $g$ is the reduced-form mapping of $\mathcal{M}$. The metric construction in the semi-latent space is simpler compared to the feature space due to the independence of its components. This independence arises from the sufficiency assumption for $\mathbf{U}_i$ and the intervention assumption for $\mathbf{V}_{\mathcal{I}}$ in the SCM. Let $(\mathcal{Q}_i, d_i)$ represent the metric space for the semi-latent space. We can define the dissimilarity function for all $Q_i$ using a product metric, similar to the Euclidean example i.e., $d(x, y) = \sqrt{\sum_{i=1}^n d_i(x_i, y_i)^2}$. We aim to ascertain the specific formulations of the causal fair metric, as delineated in Definition 4.2.

**Proposition 4.4** *Let $d : \mathcal{V} \times \mathcal{V} \to \mathbb{R}$ be a causal fair metric, then $d$ can be written as a form:*

$$d(v, v') = d_{\mathcal{X}}(\varphi_{\mathcal{X}}(v), \varphi_{\mathcal{X}}(v')), \tag{2}$$

*where $\varphi_{\mathcal{X}}(v) = P_{\mathcal{X}}(\varphi(v))$, $\varphi$ is the mapping from feature space to semi-latent space, $P_{\mathcal{X}}$ is a projection on the non-sensitive subspace of exogenous space, and $d_{\mathcal{X}}$ represents the metric defined on the non-sensitive subspace $\mathcal{U}_{\mathcal{X}}$, which exhibits continuity along its diagonal with each of its components.*

By aiding the proposition, when the semi-latent space metric is defined by an inner product, the metric takes the well-known form of a kernelized Mahalanobis distance:

$$d(v, v') = \langle (\varphi(v) - \varphi(v')), \Sigma(\varphi(v) - \varphi(v')) \rangle, \tag{3}$$

where $\Sigma$ is the projection matrix on non-sensitive exogenous space.

## 5 Protected Causal Perturbation

An adversarial perturbation ball, a fundamental concept in machine learning robustness, defines a region in the input space within which data variations do not alter the model's predicted category.

This concept evaluates the model's sensitivity to input alterations, especially under adversarial attacks designed to mislead it. Metrics are crucial in quantifying perturbations by gauging the distance between original and altered data. In this section, we extend this concept by applying fair causal metrics to define causal perturbations.

**Definition 5.1 (Protected Causal Perturbation)** *Consider an SCM $\mathcal{M}$ that includes sensitive attributes, and let $d$ represent its causal fair metric. We define the protected causal perturbation (PCP) ball with radius $\Delta$ for an instance $v$ as follows:*

$$B_{\Delta}^{PCP}(v) = \{v' \in \mathcal{V} : d(v, v') \leq \Delta\}, \tag{4}$$

*where $\Delta$ is a non-negative real number.*

We will examine how the shape of the perturbation ball changes when we add causal structures and protect sensitive features. Fig. 1 shows how the counterfactual ball evolves with these aspects. We define a closed ball $B_\Delta^{\mathcal{X}}$ in space $\mathcal{X}$ as $B_\Delta^{\mathcal{X}}(x) = \{x' \in \mathcal{X} : d_{\mathcal{X}}(x, x') \leq \Delta\}$. Equation 4 gives a simple formula: $B_\Delta^{\mathbf{PCP}}(v) = \varphi_{\mathcal{X}}^{-1}(B_\Delta^{\mathcal{X}}(\varphi(v)))$. But since $\varphi_{\mathcal{X}}$ is not bijective (because projection function $P_{\mathcal{X}}$ is not bijective) when sensitive features are present, $B_\Delta^{\mathbf{PCP}}$ and $B_\Delta^{\mathcal{X}}$ are not isomorphic. Define $B_\Delta^{\mathbf{CP}}$ as the part of $B_\Delta^{\mathbf{PCP}}$ that only includes the causal structure, leaving out the sensitive protected attributes, i.e., $B_\Delta^{\mathbf{CP}}(v) = \{v' \in \mathcal{V} : P_{\mathcal{X}}^\perp(\varphi(v)) = P_{\mathcal{X}}^\perp(\varphi(v')) \; \wedge \; \varphi_{\mathcal{X}}(v') \in B_\Delta^{\mathcal{X}}(\varphi_{\mathcal{X}}(v))\}$. We see that $B_\Delta^{\mathbf{CP}}$ is isomorphic to $B_\Delta^{\mathcal{X}}$. Thus, $B_\Delta^{\mathbf{PCP}}$ is formed by combining causal balls around each counterfactual instances of $v$.

**Proposition 5.2** *Let $B_\Delta^{PCP}(v)$ represent the PCP ball around the instance $v$ with a radius of $\Delta$. It can be decomposed as:*

$$B_\Delta^{PCP}(v) = \bigcup_{s \in \mathcal{S}} B_\Delta^{CP}(\ddot{v}_s), \tag{5}$$

*where $\mathcal{S}$ represents the level set of sensitive features (which may be continuous or discrete). $B_\Delta^{PCP}$ exhibits invariance under twins, meaning that for all $s \in \mathcal{S}$, we have $B_\Delta^{PCP}(v) = B_\Delta^{PCP}(\ddot{v}_s)$.*

The PCP definition, along with the causal fair metric property, captures the counterfactual proximity definition. The subsequent lemma demonstrates that a PCP with a diameter of 0 represents the set of twins.

**Proposition 5.3** *Let $\mathbf{S}$ denote the protected features, and let $d$ be the causal fair metric. The set of counterfactual twins corresponds to the PCP with a zero radius i.e., $\ddot{\mathbb{V}} = \lim_{\Delta \to 0} B_\Delta^{PCP}(v)$.*

# 6 Causal Fair Metric Learning

From Eq. 2, we can create a fair metric using structural equations and a metric for non-sensitive exogenous variables. This involves deriving the metric from data and dealing with unknowns such as sensitive features, the embedding function $\varphi$, and the metric for non-sensitive exogenous features. Assuming we know the sensitive features and have dissimilarity functions for each exogenous component from domain experts. With these assumptions, understanding the functional structures allows us to construct $\varphi$ and, in turn, develop a causal fair metric. This metric is fundamentally linked to counterfactuals, raising the critical question of whether it's possible to estimate counterfactuals from observational data. The below example that is adapted from Peters et al., 2017, § 6.19 investigates the possibility of this idea.

**Example 6.1** *Let $\mathcal{M}_A$ and $\mathcal{M}_B$ be two SCM with below structural equations respectively:*

$$\mathcal{M}_A = \begin{cases} V_1 := U_1, \\ V_2 := V_1(1 - U_2), \\ V_3 := \mathbb{I}_{V_1 \neq V_2}(\mathbb{I}_{U_3 > 0}V_1 + \\ \qquad \mathbb{I}_{U_3 = 0}V_2) + \mathbb{I}_{V_1 = V_2}U_3. \end{cases} \qquad \mathcal{M}_B = \begin{cases} V_1 := U_1, \\ V_2 := V_1(1 - U_2), \\ V_3 := \mathbb{I}_{V_1 \neq V_2}(\mathbb{I}_{U_3 > 0}V_1 + \\ \qquad \mathbb{I}_{U_3 = 0}V_2) + \mathbb{I}_{V_1 = V_2}(N - U_3). \end{cases}$$

*where, $U_1$ and $U_2$ have a Bernoulli distribution with a $0.5$ probability, and $U_3$ has a uniform distribution spanning from $0$ to a constant value $N$. Consider the instance $v = (1, 0, 0)$, with $V_1$ denoted as the sensitive feature. The counterfactuals for $v$ with respect to $\mathcal{M}_A$ and $\mathcal{M}_B$ are $(0, 0, 0)$ and $(0, 0, N)$, respectively.*

Both SCMs have identical causal graphs, observational distributions, and intervention distributions for all possible interventions. Thus, no randomized trials or observational data can distinguish between $\mathcal{M}_A$ and $\mathcal{M}_B$. Therefore, for counterfactual statements, additional assumptions are essential. Example 6.1 establishes the following proposition.

**Proposition 6.2 (Metric Estimation Not Guaranteed)** *If the set of descendants of intervened variables is non-empty, estimating a causal fair metric, from observational data or with a causal graph, necessitates knowledge of the true structural equations, irrespective of data quantity or type.*

Proposition 6.2 states that, in the general case where the SCM is unknown, data-driven metric learning becomes infeasible if we first attempt to estimate counterfactuals from the available sample data—even

when the causal graph can be accurately inferred. In other words, this implies that for certain SCMs, such as linear SCMs with Gaussian noise, learning the causal graph solely from sample data is impossible. Consequently, deriving a causally fair metric from the data is also unattainable.

As SCM knowledge is often elusive in practice, an alternative is estimating the causal-fair metric directly from data labeled with distances. Metric learning methods vary, including spectral, probabilistic, and deep learning. Spectral techniques use eigenvalue decomposition to represent data in a lower-dimensional space, while probabilistic methods infer a low-dimensional latent variable underlying the high-dimensional data. Both spectral and probabilistic metric learning techniques employ the generalized Mahalanobis distance, denoted as Eq. 3, with a predetermined kernel such as the Gaussian kernel or a kernel that is learned, aiming to optimize the dissimilarity matrix (Ghojogh et al., 2022).

Conversely, deep metric learning utilizes neural networks to determine the embedding function. The network aims to reduce distances between similar points while increasing distances between dissimilar ones. This approach aligns with Proposition 4.4, which asserts the existence of an embedding $\varphi_{\mathcal{X}} : \mathbb{R}^n \to \mathbb{R}^k$. The causal fair metric is defined as $d_\varphi(v, w) = d_{\mathcal{X}}(\varphi_{\mathcal{X}}(v), \varphi_{\mathcal{X}}(w))$, with $k$ indicating the dimension of the non-sensitive exogenous space. This leads to three main insights: the dimensionality of the embedding space, the guarantee of independence from coordinates in this space, and understandings about $d_{\mathcal{X}}$. These insights are crucial for creating specific deep-learning techniques for causal fair metric learning.

In designing a neural network, we focus on feed-forward neural networks with a depth of $L \geq 1$. These networks are characterized by their layer widths, denoted as $d_1, \ldots, d_L$, where $d_0 = n$ represents the input size, and $d_L = k$ represents the output size. Each layer has an element-wise activation function $\sigma_i$, which operates on $\mathbb{R}^{d_{i-1}}$ and maps to $\mathbb{R}^{d_i}$. The transformation process of the network is expressed as follows:

$$\varphi_{\mathbf{w}}(v) = \sigma_L(\mathbf{W}_L \times \sigma_{L-1}(\mathbf{W}_{L-1} \times \cdots \sigma_1(\mathbf{W}_1 \times v) \cdots)) \tag{6}$$

Consequently, the causal fair metric can be expressed as $d(v, w) = d_{\mathcal{X}}(\varphi_{\mathbf{w}}(v), \varphi_{\mathbf{w}}(w))$, where $\mathbf{W} = (\mathbf{W}_1, \ldots, \mathbf{W}_L)$ denotes a tuple of matrices. Each matrix $\mathbf{W}_i \in \mathbb{R}^{d_i \times d_{i-1}}$, and $d_{\mathcal{X}}$ is a known metric. This matrix tuple family is symbolized as $\mathcal{W}$, thus allowing the representation of the family of non-linear functions as $\Phi = \{\varphi_{\mathbf{w}} : \mathbf{W} \in \mathcal{W}\}$.

To assess how the causal fair metric affects deep learning, we need specific measures to track progress in various scenarios. Kozdoba et al. (Kozdoba & Mannor, 2021) adopted the approach from Bartlett et al. (Bartlett et al., 2017) by metric learning principles. Their results are based on the following norm definitions: The spectral norm of a matrix $W \in \mathbb{R}^{s \times t}$ is denoted as $\|W\|$. Additionally, $\|W\|_{2,1}$ is introduced as the sum of the $\ell_2$ norms of each column in matrix $W$, where $W_{\cdot,i}$ represents the $i$-th column of the matrix.

**Proposition 6.3 (Kozdoba (Kozdoba & Mannor, 2021))** *Consider a feed-forward network with L layers described in Eq. 6. Assuming that activation functions $\rho_i$ are $\lambda_i$-Lipschitz, and the feature space $\mathcal{V}$ is bounded with $\|v\|_2 \leq B$ for all $v \in \mathcal{V}$, the Rademacher complexity of $\Phi$ for a family of matrix tuples $\mathcal{W}$ is bounded as follows:*

$$\mathcal{R}(\Phi) \leq \bar{O}\left( \frac{1}{\sqrt{n}} B^2 \left( \prod_{i=1}^{n} \lambda_i \|\mathcal{W}_i\| \right)^2 \left( \sum_{i=1}^{L} \frac{\|\mathcal{W}_i\|_{2,1}^{\frac{2}{3}}}{\|\mathcal{W}_i\|^{\frac{2}{3}}} \right)^{\frac{3}{2}} \right) \tag{7}$$

*Here, $\|\mathcal{W}_i\|$ represents the supremum norm over $W$ in $\mathcal{W}$ for $W_i$, and $\|\mathcal{W}_i\|_{2,1}$ is the supremum over $W$ in $\mathcal{W}$ for $W_i$ with respect to the $\ell_{2,1}$ norm.*

Proposition 6.3 ensures that in deep metric learning, the estimator converges to the true underlying metric with a convergence rate of $\bar{O}(\frac{1}{\sqrt{n}})$, forming the core theoretical foundation of each proposed learning algorithm. Moreover, the proposition establishes that deep metric learning can effectively distinguish embeddings regardless of dimensionality or the specific metric used, as dimensions are not explicitly considered in Eq. 7. However, numerical analysis (§ 8) shows that incorporating causal fair metric assumptions improves estimation accuracy compared to general metric learning methods.

We conclude this chapter by highlighting that our approach to metric learning not only provides a framework for learning causally fair metrics but also enables metric learning even when causal graph inference is infeasible.

## 7 Causality-aware Fair Adversarial Learning

Fair adversarial learning seeks to predict the target variable accurately while maintaining fairness concerning sensitive attributes. Using the set of observations $\mathcal{D} = \{(v_i, y_i)\}_{i=1}^n$, it entails a min-max optimization problem where the model minimizes classification error and maximizes adversarial loss around each instance $v_i$.

A key insight from Prop. 6.2 is that fair adversarial learning using just observational data $\mathcal{D}$ is unattainable in causal structures, as inferring a suitable metric for assessing counterfactuals is not possible. Practically, $\Delta$ is set by varying values within a perturbation ball to include samples deemed similar. Essentially, people are learning the metric based on their experience. At best, this method estimates the upper bound of the appropriate $\Delta$, highlighting the significance of metric learning.

To initiate fair adversarial learning, the first step is to use metric estimation for constructing $B_\Delta^{\mathbf{PCP}}$, as demonstrated in the subsequent min-max adversarial learning framework:

$$\min_\psi \mathbb{E}_{(v,y) \sim \mathcal{P}_\mathcal{D}} \big[ \max_{w \in B_\Delta^{\mathbf{PCP}}(v)} \ell(h_\psi(w), y) \big],$$

where $\mathcal{P}_\mathcal{D}$ is empirical distribution corresponding to observation set $\mathcal{D}$ and $h_\psi$ is parametric model with parameter $\psi$. In Ehyaei et al. (Ehyaei et al., 2023b), the limitations of gradient descent in adversarial learning are discussed, and a superior, locally linear method named *CAPIFY* is proposed for contexts with causal structures and sensitive attributes. This approach includes integrating regularizers into the loss function, as detailed below:

$$\mathcal{R}_\Delta(v, y) = \mu_1 * \max_{s \in \mathcal{S}} \ell(h(\ddot{v}_s), y) + \mu_2 * \|\nabla_v \ell(\mathbf{CF}(v, \delta), y)_{|\delta=0}\|_* + \mu_3 * \gamma_\Delta(v, y)$$

where $\mu_i$ are regularize coefficients and the term $\gamma_\Delta(v, y)$ is obtained by:

$$\max_{\delta \in B^{\mathbf{PCP}}(\Delta)} |\ell(\mathbf{CF}(v, \delta), y) - \ell(v, y) - \delta^T \nabla_v \ell(\mathbf{CF}(v, \delta), y)_{|\delta=0}|.$$

The first part of the regularizer ensures counterfactual fairness by measuring the maximum loss of the instance label and its twins. The second and third parts assess the adversarial robustness of classifier $h$ concerning continuous features around each twin w.r.t. causal structure. Calculating these two terms requires knowledge of the causal functional structure. Without knowing the SCM, twins of an instance can be estimated using the causal fair metric introduced in Section 6. As the causal structure is unknown, by using twins, the second and third terms are estimated by $\max_{s \in \mathcal{S}} \{|\Delta^T . \nabla_v \ell(v, y)_{|\ddot{v}_s}| + \hat{\gamma}_\Delta(\ddot{v}_s, y)\}$, where $\hat{\gamma}_\Delta(v, y) = |\ell(v + \Delta, y) - \ell(v, y) - \Delta^T . \nabla_v \ell(v, y)|$.

By using the last equations, we introduce the **ECAPIFY** method, which operates without SCM knowledge and relies solely on metric learning. To train a classifier with **ECAPIFY**, the following regularizer is added to the learning loss function.

$$\hat{\mathcal{R}}_\Delta(v, y) = \max_{s \in \mathcal{S}} \{\mu_1 * \ell(h(\ddot{v}_s), y) + \mu_2 * |\Delta^T . \nabla_v \ell(\ddot{v}_s, y)| + \mu_3 * \hat{\gamma}_\Delta(\ddot{v}_s, y)\}$$

## 8 Numerical Experiments

In this section, we empirically validate the metric learning method presented in Section 6. We compare our method, which incorporates causal structure and sensitive information, to standard deep metric learning. We use deep learning to estimate the embedding function, and Siamese metric learning (Chicco, 2021) as

the baseline. Siamese networks use a neural network $\phi$ to map inputs $\mathbf{v}_1$ and $\mathbf{v}_2$ into an embedding space, minimizing $d(\phi(\mathbf{v}_1), \phi(\mathbf{v}_2))$ for similar points and maximizing it for dissimilar ones. Contrastive loss is used:

$$\mathcal{L} = y \cdot d^2 + (1 - y) \cdot \max(0, m - d)^2,$$

where $d = \|\phi(\mathbf{v}_1) - \phi(\mathbf{v}_2)\|$, $y$ indicates similarity, and $m$ is the margin. Simulations are divided into three scenarios distance-based, label-based, and triplet-based (further details in § .3).

For designing the embedding network, we use a feed-forward network with 100-node layers and *PReLU* activation. We consider two embedding layer dimensions: a known dimension and half the input size for an unknown network. We test the network's depth with either 5 or 14 hidden layers and evaluate the impact of a known metric in the exogenous space by comparing scenarios with both known and unknown metrics. We investigate the impact of assuming coordinate independence by including a decorrelation loss function (Patil & Purcell, 2022), which uses the Frobenius norm of the difference between the identity matrix and XIcor (Chatterjee, 2021), a non-parametric correlation measure, on training performance.

In our numerical experiments, a major challenge is finding well-known datasets in causal inference and metric learning. Collecting tagged data aligned with specific metrics is becoming increasingly prevalent. For example, in new Large Language Model (LLM) methodologies, tagged data plays a critical role in providing explicit feedback on which responses better meet user needs.

To address the lack of appropriate data, for our real-world datasets, Adult (Kohavi & Becker, 1996) and COMPAS (Washington, 2018), we first establish a causal structure as in Nabi et al. (Nabi & Shpitser, 2018). We also use synthetic datasets for Linear (LIN) and Non-linear (NLM) SCMs. For each SCM, we create three data scenarios using its structure. We employ the PCP ball with radii $\Delta = 0.1$, and 0.2 for contrastive label creation, generating 10,000 samples. We then assess deep metric learning across 100 iterations with varying random seeds. Note that in our study, we did not study the specifics of the embedding network architecture. Instead, we employed a straightforward feed-forward network, which is better suited for our tabular data. This choice aligns with Prop. 6.2, assisting in discerning the impacts of various assumptions.

To assess learning performance, we employ classifier metrics such as accuracy (*Acc*), Matthews correlation coefficient (*MCC*), false-negative (*FN*), and false-positive (*FP*) rates for label outputs. For embedding kernel learning, we use root mean square error (*RMSE*) and mean absolute error (*MAE*). Continuous metrics are used in both label and triplet-based kernel learning. In distance-based scenarios, label predictions are made by generating labels within the $B_\Delta^{\mathbf{PCP}}$ for uniform performance evaluation across different settings.

To evaluate the **ECAPIFY** approach, we compare it with traditional empirical risk minimization (*ERM*), Adversarial Learning (*AL*) as delineated by Madry et al. (Madry et al., 2017), and *CAPIFY*, which is recognized for its superior effectiveness in mitigating unfairness, as detailed in Ehyaei et al. (Ehyaei et al., 2023b). Our simulation settings and performance metrics mirror those in Ehyaei et al. (Ehyaei et al., 2023b). To measure unfairness we use the unfair area quantity that is defined as the below

$$\text{Unfair Area} := \mathbb{P}\big(v \in \mathcal{V} : \exists v' \text{ such that } d(v, v') \leq \Delta \text{ and } h(v) \neq h(v')\big).$$

Similarly, we define counterfactually unfair area.

$$\text{Cunterfactual Unfair Area} := \mathbb{P}\big(v \in \mathcal{V} : \exists a \in \mathcal{A} \text{ such that } h(v) \neq h(\ddot{v}_a)\big).$$

We set the perturbation radius at $\Delta = 0.01$ and report the percentages of non-robust, non-counterfactual instances, and their combination, along with accuracy. Additional simulation details are in the appendix, and our numerical analysis codes are available on GitHub.

**Results of Metric Learning**  As demonstrated in Fig. 2 and in details in Tab. 1, our simulation confirms that knowing the metric and dimensions of the embedding space improves accuracy in metric learning. Although Prop. 6.2 asserts that deep learning can converge with various layer sizes without embedding space knowledge, we show that additional information significantly enhances results, particularly in triplet-based scenarios.

| | | Real-World Data | | | | | | | | Synthetic Data | | | | | | | |
| --- | --- | --- | --- | --- | --- | --- | --- | --- | --- | --- | --- | --- | --- | --- | --- | --- | --- |
| | | **Adult** | | | | **COMPAS** | | | | **Lin** | | | | **NLM** | | | |
| Δ | Loss Function | Acc↑ | FN↓ | MAE↓ | RMSE↓ | Acc | FN | MAE | RMSE | Acc | FN | MAE | RMSE | Acc | FN | MAE | RMSE |
| | Distance-based | **0.948** | 0.036 | **0.029** | **0.044** | **0.989** | 0.008 | **0.006** | **0.009** | **0.992** | 0.007 | **0.004** | **0.005** | **0.976** | 0.018 | **0.007** | **0.013** |
| 0.10 | Label-based | 0.822 | **0.003** | 0.098 | 0.134 | 0.842 | **0.000** | 0.098 | 0.130 | 0.829 | **0.000** | 0.096 | 0.129 | 0.843 | **0.000** | 0.095 | 0.126 |
| | Triplet-based | 0.619 | 0.202 | 0.173 | 0.228 | 0.614 | 0.206 | 0.177 | 0.233 | 0.637 | 0.219 | 0.177 | 0.233 | 0.738 | 0.200 | 0.174 | 0.230 |
| | Distance-based | **0.955** | 0.033 | **0.050** | **0.078** | **0.988** | 0.010 | **0.010** | **0.017** | **0.990** | 0.008 | **0.006** | **0.008** | **0.991** | 0.007 | **0.007** | **0.013** |
| 0.20 | Label-based | 0.825 | **0.002** | 0.192 | 0.266 | 0.850 | **0.000** | 0.193 | 0.257 | 0.838 | **0.000** | 0.195 | 0.259 | 0.852 | **0.000** | 0.189 | 0.252 |
| | Triplet-based | 0.805 | 0.187 | 0.346 | 0.457 | 0.841 | 0.073 | 0.353 | 0.466 | 0.663 | 0.210 | 0.354 | 0.467 | 0.743 | 0.150 | 0.348 | 0.459 |

Table 1: The table shows results of a numerical experiment comparing different learning scenarios, evaluated by accuracy (**Acc** - higher is better), false negative error (**FN** - lower is better), root mean square error (**RMSE** - lower is better), and mean average error (**MAE** - lower is better). The best scenario for each dataset and perturbation radius is in bold. XIcor correlation loss function and a 5-layer embedding network are used. To show variation, we included error bars in Figures 2–4 and detailed measures in Tables 2–4 in the appendix. This table shows our framework effectively estimates causal fair metrics, achieving high confidence across datasets, perturbation radii, and tagged data scenarios.

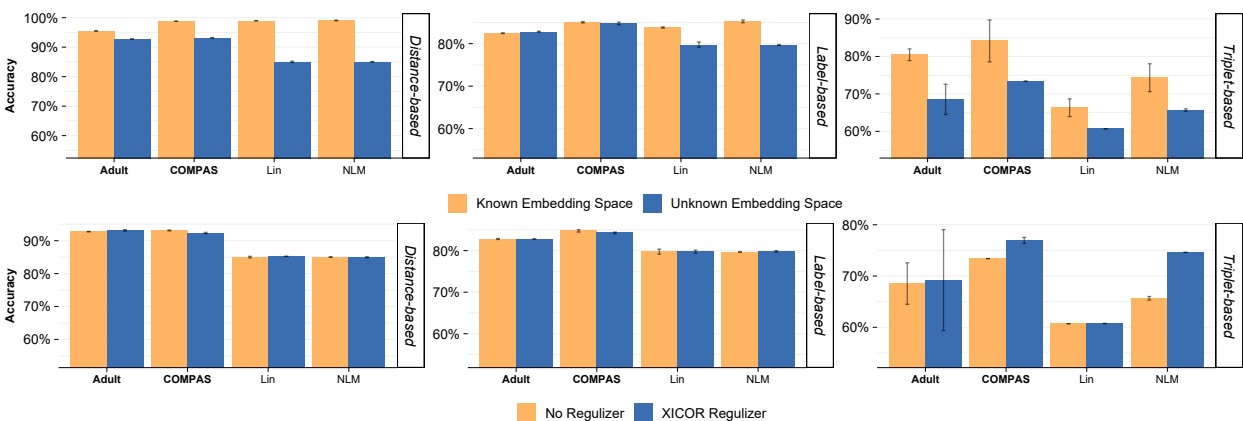

Figure 2: This figure demonstrates the effect of causal metric assumptions on the accuracy of deep metric models: (up) Accuracy performance comparison based on embedding layer sizes and embedding space metric knowledge shows improved prediction accuracy. (down) In simpler models, the network efficiently learns embedding space properties. However, with less precise metric data, as in Triplet-based scenarios, adding decorrelation methods boosts accuracy.

Fig. 2 shows that embedding learning works well in distance-based and label-based scenarios, where adding the decorrelation loss function does not make a big difference. But in the triplet scenario, where only metric relations are known, this loss function improves results.

To find the optimal configurations for embedding network layers, we ran experiments with various depths of networks. We find that five layers network is ideal for label-based and distance-based scenarios, whereas triplet-based scenarios perform better with deeper network structures. Further results are available in the Appendix (see Tab. 4). To summarize simulation results, analysis of various learning methodologies shows that distance-based metric learning is most effective when precise distance-based data is available. However, in practical situations, this ideal may not be achievable. In such cases, the label-based method becomes a viable alternative for metric approximation. This method's accuracy improves when embedding space dimensions and metric information is combined, as Tab. 1 supports. The label-based approach also has a lower false negative rate compared to other methods, making it effective in approximating the true metric. This is particularly useful in scenarios requiring fair metrics, like robust learning, as it helps maintain robustness criteria and builds a stronger model. When label data is unavailable, the triplet method, enhanced with a decorrelation loss function and deeper networks, effectively deduces the embedding function.

**Results of ECAPIFY.** In Fig. 3, our simulation results, utilizing both real-world and synthetic datasets, show that ECAPIFY, which is equipped with metric learning and not necessitating knowledge of SCM,

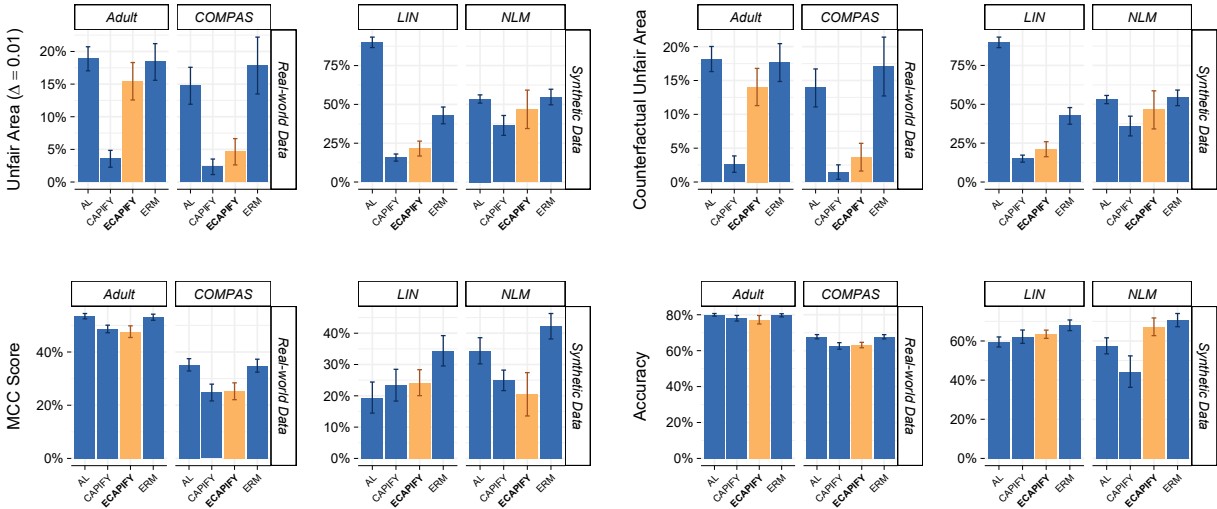

Figure 3: Presents the results of our numerical experiment, evaluating ECAPIFY's performance across various models and datasets. (Top left) Bar plot comparing models using unfair area percentage (lower is better) at $\Delta = .01$. (Top right) Counterfactual unfair area percentage (lower is better). (Bottom left) Matthews correlation coefficient illustrating classifier performance (higher is better). (Bottom right) Bar plot contrasting methods by prediction performance (higher is better). CAPIFY performs better because it assumes knowledge of the causal metric, whereas ECAPIFY first learns the metric before fitting the model.

yields results akin to CAPIFY (regarded as an oracle), achieving similar effectiveness in diminishing unfair areas with $\Delta = 0.01$. Additionally, there is a notable performance improvement compared to both ERM and AL, which concentrate on adversarial training, while still preserving high prediction accuracy. A key aspect of ECAPIFY is that, despite Prop. 6.2 highlighting the impracticality of adversarial training with only observational data, ECAPIFY offers a feasible approach using empirical data. Essentially, without knowledge of the SCM, estimating a causal fair metric is crucial for adversarial learning.

## 9 Discussion and Future Work

In this study, we introduce the concept of a causal fair metric and propose a protected causal perturbation to integrate individual fairness, adversarial robustness, and causality. By leveraging causal structures and protecting sensitive attributes, our approach enables the training of a causally aware, fair, and robust classifier without relying on a structural causal model (SCM). Instead, we utilize metric learning to balance individual fairness and adversarial robustness.

Our approach prioritizes fairness, robustness, and causality, aligning with responsible AI principles. However, it inherits vulnerabilities from underlying models, such as issues with privacy, explainability, and safety, requiring careful consideration in high-stakes applications. We emphasize that this work serves as a proof of concept, encouraging further exploration and collaboration. Although our method demonstrates promising results, it also highlights key limitations and avenues for future exploration.

Our methodology assumes an additive noise model, which may not fully encapsulate the complexity of real-world causal relationships, complicating additive interventions in general SCMs. Additionally, despite similarities to existing metric learning methods, our approach lacks theoretical guarantees for estimator performance and faces challenges such as local minima. Future work aims to address these issues by imposing constraints on the causal fair metric structure and developing explicit convergence theorems. Another significant challenge is the scarcity of real-world datasets suited for metric learning with causal structures. Furthermore, our proposed notion of protected causal perturbation extends to other domains within causal machine learning, such as algorithmic recourse, causal bandits, and reinforcement learning, offering a broader framework for analyzing fairness and robustness.

## Acknowledgments

The authors thank the Max Planck Institute for Intelligent Systems, Tübingen AI Center, for supporting this project. Partial funding support was also provided by the Canada CIFAR AI Chair program.

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

| Symbol | Notion |
|---|---|
| SCM | Structural causal model |
| $d(v_i, v_j)$ | Fair distance metric between $v_i$ and $v_j$ |
| $\mathcal{M}$ | Structural causal model |
| $\mathbf{V}$ | Feature or endogenous space includes $n$ random variables $\{\mathbf{V}_i\}_{i=1}^n$ |
| $\mathbf{U}$ | noise or exogenous space includes $n$ random variables $\{\mathbf{U}_i\}_{i=1}^n$ |
| $\mathcal{G}$ | Causal graph |
| $\mathbb{F}$ | Set of structural equations $f_i$ |
| $\mathbb{P}_{\mathbf{U}}$ | Exogenous probability distribution |
| $g : \mathcal{U} \to \mathcal{V}$ | reduced-form mapping from $\mathcal{U}$ to $\mathcal{V}$ |
| $\mathbb{P}_{\mathbf{V}}$ | Feature probability distribution |
| ANM | Additive noise model |
| $\mathcal{M}^{do(\mathbf{V}_\mathcal{I} := \theta)})$ | Hard intervention respect to $\mathcal{I}$ subset of feature |
| $\mathcal{M}^{do(\mathbf{V}_\mathcal{I} += \delta)}$ | Additive intervention |
| $\mathbf{CF}(v, \theta)$ | counterfactual instance respect to hard intervention |
| $g^\theta$ | Altered reduced-form mapping w.r.t. $\mathcal{M}^{do(\mathbf{V}_\mathcal{I} := \theta)}$ |
| $\mathbf{S}$ | Sensitive attribute |
| $\mathcal{S}$ | The level sets of sensitive attribute |
| $\ddot{v}_s = \mathrm{CF}(v, s)$ | Counterfactual twin for $\mathbf{S} = s$ |
| $\ddot{\mathbf{v}}$ | Set of all twins |
| $d_\mathcal{X}$ | Metric on feature space |
| $d_\mathcal{Y}$ | Metric on label space |
| $h : \mathcal{X} \to \mathcal{Y}$ | Classifier function |
| $\mathrm{CF}(v, \delta)$ | counterfactual for additive noise interventions |
| $d : \mathcal{V} \times \mathcal{V} \to \mathbb{R}_{\geq 0}$ | Causal fair metric |
| $\delta_{|\mathcal{I}} = 0$ | Causal perturbation over non-sensitive part |
| $\mathcal{B}(p)$ | Bernoulli distribution |
| $\mathcal{Q}$ | Semi-latent space |
| $\varphi : \mathcal{V} \to \mathcal{Q}$ | embedding map from $v$ to the semi-latent space |
| $\varphi^{-1}$ | Inverse of embedding map $\varphi$ |
| $(\mathcal{Q}_i, d_i)$ | metric space for each semi-latent space component |
| $\mathcal{X}$ | Non-sensitive part of semi-latent space |
| $d_\mathcal{X}$ | Metric that is define on $\mathcal{X}$ |
| $P_\mathcal{X}$ | Projects semi-latent spate to $\mathcal{X}$ |

| Symbol | Notion |
|---|---|
| $\varphi_\mathcal{X} = P_\mathcal{X}(\varphi)$ | Combination of embedding and projection on $\mathcal{X}$ |
| PCP | Protected causal perturbation |
| $B_\Delta^{\mathbf{PCP}}(v)$ | PCP ball for instance $v$ with radii $\Delta$ |
| $B_\Delta^\mathcal{X}$ | simple closed ball with radii $\Delta$ in $\mathcal{X}$ |
| $B_\Delta^{\mathbf{CP}}$ | Non-sensitive part of $B_\Delta^{\mathbf{PCP}}$ ball |
| $\sigma_i$ | activation function on layer $i$-th |
| $\varphi_{\mathbf{w}}$ | Neural net embedding function |
| $\mathbf{W}$ | Matrix parameters of neural net |
| $\mathcal{W}$ | Parameter space for tuples $\mathbf{W}$ |
| $\Phi$ | Space of all embedding functions |
| $\|W\|$ | Spectral norm on Matrix |
| $\|W\|_{2,1}$ | Sum of the $\ell_2$ norms of each column in matrix $W$ |
| $\mathcal{R}(\Phi)$ | Rademacher complexity of $\Phi$ |
| $\mathcal{D}$ | $\{(v_i, y_i)\}_{i=1}^n$ set of observational data |
| $\mathcal{P}_\mathcal{D}$ | Observational probability |
| $\ell$ | Learning loss function |
| $\mathcal{R}_\Delta$ | CAPIFY regularizer |
| $\hat{\mathcal{R}}_\Delta$ | ECAPIFY regularizer |
| $L_\delta$ | Huber loss function |
| $[.]_+$ | Standard Hinge loss function |
| Prop. | Proposition |
| Fig. | Figure |
| Eq. | Equation |

### .1 Additional Background

**Definition .1 ((Pseudo-) Metric Space)** *A metric space $(X, d)$ is defined as a set $X$ accompanied by a non-negative real-valued function $d : X \times X \longrightarrow \mathbb{R}_{\geq 0}$, which is referred to as a metric. This metric function $d$ adheres to the subsequent properties for any $x, y, z \in X$:*

- **Non-negativity:** *$d(x, y) \geq 0$, and $d(x, y) = 0$ if and only if $x = y$.*

- **Symmetry:** *$d(x, y) = d(y, x)$.*

- **Triangle inequality:** *$d(x, z) \leq d(x, y) + d(y, z)$.*

*When the positivity condition, i.e., $d(x, y) = 0$ if and only if $x = y$ is relaxed, the $d$ is called pseudometric (or semi-metric).*

**Definition .2 (The pull-back & push-forward metric)** *let $f : \mathcal{U} \to \mathcal{V}$ be a mapping between the metric spaces $(\mathcal{U}, d_{\mathcal{U}})$ and $(\mathcal{V}, d_{\mathcal{V}})$. The push-forward metric $d$ induced by the function $f$ is defined as:*

$$d(u_1, u_2) = d_{\mathcal{V}}(f(u_1), f(u_2)); \quad u_1, u_2 \in \mathcal{U}$$

*Similarly, the pull-back metric on the space $\mathcal{U}$ is defined as:*

$$d(v_1, v_2) = d_{\mathcal{U}}(f^{-1}(v_1), f^{-1}(v_2)); \quad v_1, v_2 \in \mathcal{V}$$

*These definitions allow us to relate distances in $\mathcal{U}$ and $\mathcal{V}$ via the mapping $f$ and its inverse $f^{-1}$.*

**Definition .3 (Huber Loss)** *For a given predicted value $\hat{y}$ and true target value $y$, the Huber loss function is defined as:*

$$L_\delta(\hat{y}, y) = \begin{cases} \frac{1}{2}(\hat{y} - y)^2, & \text{if } |\hat{y} - y| \leq \delta \\ \delta|\hat{y} - y| - \frac{1}{2}\delta^2, & \text{otherwise} \end{cases}$$

*where $\hat{y}$ is the predicted value, $y$ is the true target value, and $\delta$ is a positive constant that determines the threshold for switching from quadratic loss (L2) to linear loss (L1).*

### .2 Proofs

**Proposition 4.4.**

Let's consider a causal fair metric denoted as $d : \mathcal{V} \times \mathcal{V} \to \mathbb{R}$, with an associated embedding $\varphi : \mathcal{V} \to \mathcal{Q}$, mapping from the feature space to a semi-latent space. We define $d^*$ as the pull-back metric of $d$ onto $\mathcal{Q}$:

$$d^*(q_1, q_2) = d(\varphi^{-1}(q_1), \varphi^{-1}(q_2))$$

$d^*$ possesses metric properties, and we aim to elucidate which properties it inherits from Definition 4.2. We consider a decomposition of $\mathcal{Q}$ into $\mathcal{S} \times \mathcal{X}$, and let $q = \varphi^{-1}(v)$, where $v \in \mathcal{V}$. Utilizing this decomposition, we express $q$ as $(s, x)$. Property $(i)$ of the causal fair metric implies:

$$d(v, \ddot{v}_{s'}) = d^*((s, x), (s', x)) = 0 \qquad \forall s' \in \mathcal{S}$$

This property implies that $d^*$ is invariant to the sensitive part $\mathcal{S}$. To demonstrate this, we assert that for any two points $q_1 = (s_1, x_1)$ and $q_2 = (s_2, x_2)$, along with an arbitrary $s_0 \in \mathcal{S}$, the following equality holds:

$$d^*((s_1, x_1), (s_2, x_2)) = d^*((s_0, x_1), (s_0, x_2))$$

By utilizing the triangle property of $d^*$, we can establish:

$$d^*((s_1, x_1),(s_2, x_2)) \le d^*((s_0, x_1), (s_2, x_2)) + d^*((s_1, x_1), (s_0, x_1)) \Longrightarrow$$
$$d^*((s_1, x_1), (s_2, x_2)) \le d^*((s_0, x_1), (s_2, x_2))$$

The distance $d^*((s_1, x_1), (s_0, x_1))$ is zero due to the first property. Similarly, it can be shown that:

$$d^*((s_0, x_1),(s_2, x_2)) \le d^*((s_1, x_1), (s_2, x_2)) + d^*((s_1, x_1), (s_0, x_1)) \Longrightarrow$$
$$d^*((s_0, x_1), (s_2, x_2)) \le d^*((s_1, x_1), (s_2, x_2))$$

it concludes that:

$$d^*((s_0, x_1), (s_2, x_2)) = d^*((s_1, x_1), (s_2, x_2))$$

similarly, we can show:

$$d^*((s_0, x_1), (s_2, x_2)) = d^*((s_0, x_1), (s_0, x_2))$$

This last equation implies that $d^*$ is invariant to the sensitive subspace. If we consider $d_{\mathcal{X}}$ as the induced metric of $d^*$ on the sensitive subspace $\mathcal{X}$, then we can express:

$$d^*((s_1, x_1), (s_2, x_2)) = d_{\mathcal{X}}(x_1, x_2)$$

The second property of Def. 4.2 can be expressed in a simplified form based on $d_{\mathcal{X}}$. It implies that for every $x \in \mathcal{X}$, the distance $d_{\mathcal{X}}(x, x + \delta)$, where $\delta \in \mathbb{R}^{\dim(\mathcal{X})}$, is continuous with respect to $\delta$. This continuity implies that $d_{\mathcal{X}}$ is continuous along each component on its diagonal, i.e., $(x, x)$.

Finally, if we replace $x$ with $P_{\mathcal{X}}(\varphi(v))$, where $P_{\mathcal{X}}$ is the projection operator onto the subspace $\mathcal{X}$ within $\mathcal{Q}$, we obtain:

$$d(v, w) = d_{\mathcal{X}}(P_{\mathcal{X}}(\varphi(v)), P_{\mathcal{X}}(\varphi(w)))$$

This equation completes the proof.

**Proposition 5.2.**

The proof is straightforward when we write out the definitions. Let $\varphi(v) = (s, x)$ represent the embedding of the variable $v$ in the semi-latent space. To begin, we can demonstrate how the semi-latent space enables us to describe the counterfactual of instance $v$ concerning the hard action $do(\mathbf{S}{:=}s')$ as follows:

$$\varphi^{-1}(\varphi(v) \odot_I s') = \varphi^{-1}((s, x) \odot_I s') = \varphi^{-1}((s', x)) = \mathbf{CF}(v, do(\mathbf{S}{:=}s')) = \ddot{v}_{s'}$$

In the above Equation, we use the symbol $v \odot_I \theta$ to represent a masking operator that modifies the values of the entries corresponding to set $I$ in vector $v$ by replacing them with $\theta$. The validity of the last line in Equation .2 is based on the definition of the semi-latent space embedding.

By the definition 5.1, the $B_{\Delta}^{\mathbf{PCP}}(v)$ is equal to:

$$B_{\Delta}^{\mathbf{PCP}}(v) = \{v' \in \mathcal{V} : d(v, v') \le \Delta\} = \{v' \in \mathcal{V} : d_{\mathcal{X}}(P_{\mathcal{X}}(\varphi(v)), P_{\mathcal{X}}(\varphi(v'))) \le \Delta\} =$$
$$\{v' \in \mathcal{V} : d_{\mathcal{X}}(x, x') \le \Delta\} = \bigcup_{s \in \mathcal{S}} \{v' \in \mathcal{V} : \varphi(v') = (s, x') \wedge d_{\mathcal{X}}(x, x') \le \Delta\} =$$
$$\bigcup_{s \in \mathcal{S}} \{v' \in \mathcal{V} : P_{\mathcal{X}}^{\perp}(\varphi(v')) = s \ \wedge \ d_{\mathcal{X}}(\varphi_{\mathcal{X}}(\ddot{v}_s), \varphi_{\mathcal{X}}(v')) \le \Delta\} =$$
$$\bigcup_{s \in \mathcal{S}} \{v' \in \mathcal{V} : P_{\mathcal{X}}^{\perp}(\varphi(\ddot{v}_s)) = P_{\mathcal{X}}^{\perp}(\varphi(v')) \ \wedge \ \varphi_{\mathcal{X}}(v') \in B_{\Delta}^{\mathcal{X}}(\varphi_{\mathcal{X}}(\ddot{v}_s))\} =$$
$$\bigcup_{s \in \mathcal{S}} B_{\Delta}^{\mathbf{CP}}(\ddot{v}_s)$$

The last equation completes the proof.

**Proposition 5.3.**

To present the result, we must first prove the following lemma:

**Lemma .4** *Let $d$ be a causal metric, and let $d_{\mathcal{X}}$ be the corresponding embedding metric on the non-sensitive part of the exogenous space. For the closed ball $B_{\Delta}^{\mathcal{X}}$, we have:*

$$\lim_{\Delta \to 0} B_{\Delta}^{\mathcal{X}}(x) = x$$

**Proof .5** *We establish the aforementioned lemma through a proof by contradiction. Let us assume that there exists another point, denoted as $x' \neq x$, within the set $\lim_{\Delta \to 0} B_{\Delta}^{\mathcal{X}}(x)$. Consequently, we have $d_{\mathcal{X}}(x, x') = 0$. If we consider $v' = \varphi^{-1}((s, x'))$, then for $v'$, we have $d(v, v') = 0$, since $v' \notin \{\ddot{v}_s\}$. However, this contradicts the property inherent to one of the causal fair metrics.*

By utilizing the above lemma and Prop. 5.2, we can represent the result as follows:

$$B_0^{\mathbf{PCP}}(v) = \lim_{\Delta \to 0} B_{\Delta}^{\mathbf{PCP}}(v) = \lim_{\Delta \to 0} \bigcup_{s \in \mathcal{S}} B_{\Delta}^{\mathbf{CP}}(\ddot{v}_s) = \bigcup_{s \in \mathcal{S}} \lim_{\Delta \to 0} B_{\Delta}^{\mathbf{CP}}(\ddot{v}_s) =$$

$$\bigcup_{s \in \mathcal{S}} \lim_{\Delta \to 0} \{v' \in \mathcal{V} : \varphi(v') = (s', x') \wedge s' = s \ \wedge \ x' \in B_{\Delta}^{\mathcal{X}}(x)\} =$$

$$\bigcup_{s \in \mathcal{S}} \{v' \in \mathcal{V} : \varphi(v') = (s', x') \wedge s' = s \ \wedge \ x' \in \lim_{\Delta \to 0} B_{\Delta}^{\mathcal{X}}(x)\} =$$

$$\bigcup_{s \in \mathcal{S}} \{v' \in \mathcal{V} : \varphi(v') = (s, x)\} = \bigcup_{s \in \mathcal{S}} \ddot{v}_s$$

## .3 Simulation Scenarios

- **Distance-based**: Utilizing distance-tagged triplets $(v_i, v_i', d_i)$, where $d_i$ indicates the non-negative real number $d(v_i, v_i')$ as a distance. We also apply the Huber function $\ell(v_i, v_i', d_i) = L_\delta(d_i, d(\varphi(v_i), \varphi(v_i')))$ for learning loss.

- **Label-based**: Utilizing a Siamese network (Chicco, 2021) with a contrastive loss for triplets $(v_i, v_i', y_i)$, where $y_i \in \{0, 1\}$ indicates proximity between points, and the loss function $\ell(v_i, v_i', y_i)$ equals to $(1 - y_i)d(\varphi(v_i), \varphi(v_i')) + y_i[-d(\varphi(v_i), \varphi(v_i')) + m]_+$, here $m > 0$ is the marginal and $[.]_+ := \max(., 0)$ is standard Hinge loss.

- **Triplet-based**: In this approach, tuples $(v_i^1, v_i^2, v_i^3, y_i)$ are considered, where $y_i$ denotes the closeness of $v_i^1$ to $v_i^2$ compared to $v_i^1$ and $v_i^3$. Embedding is trained using a Siamese network with the triplet loss function $\ell(v_i^1, v_i^2, v_i^3, y_i) = [d(\varphi(v_i^1), \varphi(v_i^2)) - d(\varphi(v_i^1), \varphi(v_i^3)) + m]_+$.

## .4 Synthetic Data Models

In the § 8, we detail the structural equations employed to formulate the SCMs for both LIN and NLM models. The protected feature, denoted as $\mathbf{S}$, and the non-sensitive variables represented by $\mathbf{X}_i$ are derived based on the subsequent structural equations:

- linear SCM (LIN):

$$\mathbb{F} = \begin{cases} S := U_S, & U_S \sim \mathcal{B}(0.5) \\ X_1 := 2S + U_1, & U_1 \sim \mathcal{N}(0, 1) \\ X_2 := S - X_1 + U_2, & U_2 \sim \mathcal{N}(0, 1) \end{cases}$$

- Non-linear Model (NLM)

$$\mathbb{F} = \begin{cases} S := U_S, & U_S \sim \mathcal{B}(0.5) \\ X_1 := 2S^2 + U_1, & U_1 \sim \mathcal{N}(0, 1) \\ X_2 := S - X_1^2 + U_2, & U_2 \sim \mathcal{N}(0, 1) \end{cases}$$

Where $\mathcal{B}(p)$ represents Bernoulli random variables characterized by a probability $p$, and $\mathcal{N}(\mu, \sigma^2)$ denotes normal random variables, which are defined by a mean of $\mu$ and a variance of $\sigma^2$.

## .5 Real-World Data

In our study, we employed the Adult (Kohavi & Becker, 1996) and COMPAS (Washington, 2018) datasets, constructing an SCM from the causal graph by Nabi (Nabi & Shpitser, 2018). For the Adult dataset, we considered features like **sex**, **age**, and **education-num**, with sex as a sensitive attribute. For COMPAS, features included **age**, **race**, and **priors count**, with sex as the sensitive attribute.

## .6 Hyperparameter Tuning

In our experimental setup, we generated 10,000 samples for each SCM model. The data was divided into batches of 1,000, and the learning process spanned 100 epochs. The coefficient of the decorrelation regularizer was set to 0.1. Furthermore, in the contrastive label-based scenario, the margin was set equal to the radius of the experiment, while in the triplet-based scenario, the margin was set to zero to have more sensitivity for metric learning.

## .7 Training Methods

In our study, we train decision-making classifiers, denoted as $h(x)$, using various training objectives:

- **Empirical Risk Minimization (ERM)**: Minimizes expected risk for classifier parameters $\psi$, defined as:
$$\min_{\psi} \mathbb{E}_{(v,y) \sim \mathcal{P}_{\mathcal{D}}}[\ell(h_{\psi}(v), y)]$$

- **Adversarial Learning (AL)**: Trains the model against adversarial perturbation:
$$\min_{\psi} \mathbb{E}_{(v,y) \sim \mathcal{P}_{\mathcal{D}}}[\max_{\delta \in B_{\Delta}(v)} \ell(h_{\psi}(v + \delta), y)]$$

- **CAPIFY**: Combines locally linear (Qin et al., 2019) method principles with known CAP as a perturbation attack:
$$\min_{\psi} \mathbb{E}_{(v,y) \sim \mathcal{P}_{\mathcal{D}}}[\ell(h_{\psi}(x), y) + \mu_1 * \max_{s \in \mathcal{S}} \ell(h(\ddot{v}_s), y) + \mu_2 * \gamma(\Delta, v) + \mu_3 * \|\nabla_v^{\mathcal{X}} f(v)\|_*]$$

- **ECAPIFY**: Combines LLR method principles with CAP as a perturbation attack:
$$\min_{\psi} \mathbb{E}_{(v,y) \sim \mathcal{P}_{\mathcal{D}}}[\ell(h_{\psi}(x), y) + \max_{s \in \mathcal{S}}\{\mu_1 * \ell(h(\ddot{v}_s), y) + \mu_2 * |\Delta^T . \nabla_v \ell(\ddot{v}_s, y)| + \mu_3 * \hat{\gamma}_{\Delta}(\ddot{v}_s, y)\}]$$

We utilize binary cross-entropy loss as our loss function $\ell$.

## .8 Metrics

We use different metrics to evaluate trainers' performance in terms of accuracy, CAPI fairness (Ehyaei et al., 2023b), counterfactual fairness, and adversarial robustness:

- **Acc**: Classifier accuracy, expressed as a percentage.

- **M**: The Matthews Correlation Coefficient (MCC) for binary classification quality. It ranges from $-1$ (perfect inverse prediction) to $+1$ (perfect prediction), with 0 indicating random prediction. Formula:
$$\frac{(TP \times TN - FP \times FN)}{\sqrt{(TP + FP)(TP + FN)(TN + FP)(TN + FN)}}$$
Where TP, TN, FP, and FN are True Positives, True Negatives, False Positives, and False Negatives, respectively.

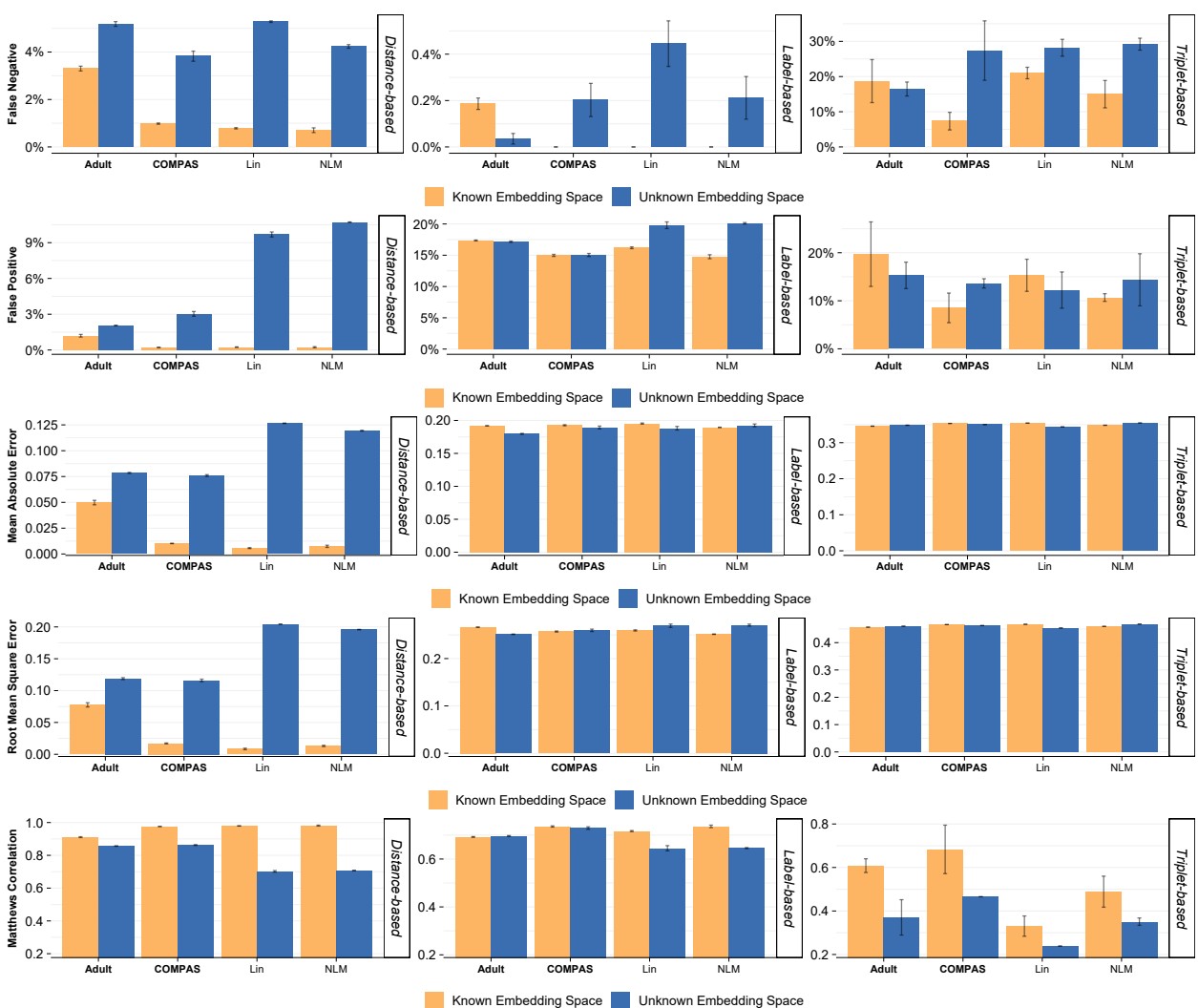

Figure 4: The performance metric of the learning scenario with varying knowledge regarding the embedding layer size.

- **MAE**: Mean Absolute Error is calculated as MAE $= \frac{1}{n}\sum_{i=1}^{n}|y_i - \hat{y}_i|$.

- **RMSE**: Root Mean Square Error with formulas: $= \sqrt{\frac{1}{n}\sum_{i=1}^{n}(y_i - \hat{y}_i)^2}$.

- **UnfairArea**: Proportion of data points within the unfair area of radius $\Delta$ as defined in Ehyaei (Ehyaei et al., 2023b).

- **Non − RobustArea**: Fraction of non-robust data points to adversarial perturbation within radius $\Delta$, equivalent to the unfair area in the absence of a sensitive attribute.

- **CounterfactualUnfairArea**: Percentage of data points showing counterfactual unfairness, analogous to the unfair area when perturbation radius is zero.

## .9 Additional Numerical Results

In the subsequent tables and figures, additional numerical analysis results are presented to support the assertions of this study. Their explanations can be found in § 8.

| Loss Function | Performance Metric | Embedding Layer Dimension | |
|---|---|---|---|
| | | Known Embedding Space | Unknown Embedding Space |
| Distance-based | Accuracy ↑ | 0.983 ± 0.014 | 0.882 ± 0.038 |
| | False Negative ↓ | 0.013 ± 0.01 | 0.043 ± 0.005 |
| | False Positive ↓ | 0.005 ± 0.004 | 0.075 ± 0.039 |
| | Matthews Correlation ↑ | 0.965 ± 0.027 | 0.766 ± 0.073 |
| | Mean Absolute Error ↓ | 0.016 ± 0.017 | 0.105 ± 0.023 |
| | Root Mean Square Error ↓ | 0.025 ± 0.027 | 0.17 ± 0.041 |
| Label-based | Accuracy ↑ | 0.845 ± 0.012 | 0.812 ± 0.021 |
| | False Negative ↓ | 0 ± 0.001 | 0.003 ± 0.003 |
| | False Positive ↓ | 0.155 ± 0.012 | 0.186 ± 0.019 |
| | Matthews Correlation ↑ | 0.725 ± 0.019 | 0.67 ± 0.036 |
| | Mean Absolute Error ↓ | 0.192 ± 0.003 | 0.19 ± 0.003 |
| | Root Mean Square Error ↓ | 0.257 ± 0.006 | 0.266 ± 0.009 |
| Triplet-based | Accuracy ↑ | 0.763 ± 0.08 | 0.708 ± 0.088 |
| | False Negative ↓ | 0.226 ± 0.154 | 0.292 ± 0.133 |
| | False Positive ↓ | 0.109 ± 0.076 | 0.107 ± 0.063 |
| | Matthews Correlation ↑ | 0.529 ± 0.157 | 0.415 ± 0.176 |
| | Mean Absolute Error ↓ | 0.351 ± 0.004 | 0.35 ± 0.004 |
| | Root Mean Square Error ↓ | 0.463 ± 0.004 | 0.462 ± 0.004 |

Table 2: The table displays the average performance metrics for comparing scenarios with knowledge of embedding dimensions and their corresponding metrics against scenarios with no knowledge of the embedding space. Green cell highlights denote superior performance, while smaller values indicate the standard deviation of the estimations.

| Loss Function | Performance Metric | Decorrelation Regularizer Function | |
|---|---|---|---|
| | | - | XICOR |
| Distance-based | Accuracy ↑ | 0.984 ± 0.012 | 0.983 ± 0.014 |
| | False Negative ↓ | 0.012 ± 0.008 | 0.013 ± 0.01 |
| | False Positive ↓ | 0.004 ± 0.004 | 0.005 ± 0.004 |
| | Matthews Correlation ↑ | 0.969 ± 0.023 | 0.965 ± 0.027 |
| | Mean Absolute Error ↓ | 0.016 ± 0.017 | 0.016 ± 0.017 |
| | Root Mean Square Error ↓ | 0.024 ± 0.028 | 0.025 ± 0.027 |
| Label-based | Accuracy ↑ | 0.843 ± 0.017 | 0.845 ± 0.012 |
| | False Negative ↓ | 0 ± 0.001 | 0 ± 0.001 |
| | False Positive ↓ | 0.156 ± 0.017 | 0.155 ± 0.012 |
| | Matthews Correlation ↑ | 0.721 ± 0.028 | 0.725 ± 0.019 |
| | Mean Absolute Error ↓ | 0.19 ± 0.008 | 0.192 ± 0.003 |
| | Root Mean Square Error ↓ | 0.256 ± 0.008 | 0.257 ± 0.006 |
| Triplet-based | Accuracy ↑ | 0.669 ± 0.032 | 0.763 ± 0.08 |
| | False Negative ↓ | 0.318 ± 0.146 | 0.226 ± 0.154 |
| | False Positive ↓ | 0.117 ± 0.09 | 0.109 ± 0.076 |
| | Matthews Correlation ↓↑ | 0.339 ± 0.063 | 0.529 ± 0.157 |
| | Mean Absolute Error ↓ | 0.351 ± 0.002 | 0.351 ± 0.004 |
| | Root Mean Square Error ↓ | 0.464 ± 0.003 | 0.463 ± 0.004 |

Table 3: The table displays average performance metrics for various scenarios, considering the presence of different decorrelation policies. Green cells highlight the best performance, while smaller values represent standard deviations of the estimates.

| Loss Function | Performance Metric | Network Layer | |
| --- | --- | --- | --- |
| | | CIFNet 14 Layers | CIFNet 5 Layers |
| Distance-based | Accuracy ↑ | $0.846 \pm 0.066$ | $0.924 \pm 0.091$ |
| | False Negative ↓ | $0.078 \pm 0.039$ | $0.047 \pm 0.053$ |
| | False Positive ↓ | $0.076 \pm 0.035$ | $0.029 \pm 0.039$ |
| | Matthews Correlation ↑ | $0.693 \pm 0.131$ | $0.849 \pm 0.182$ |
| | Mean Absolute Error ↓ | $0.07 \pm 0.049$ | $0.029 \pm 0.038$ |
| | Root Mean Square Error ↓ | $0.104 \pm 0.07$ | $0.044 \pm 0.056$ |
| Label-based | Accuracy ↑ | $0.799 \pm 0.028$ | $0.819 \pm 0.032$ |
| | False Negative ↓ | $0.008 \pm 0.009$ | $0.005 \pm 0.012$ |
| | False Positive ↓ | $0.192 \pm 0.027$ | $0.176 \pm 0.024$ |
| | Matthews Correlation ↑ | $0.644 \pm 0.049$ | $0.68 \pm 0.06$ |
| | Mean Absolute Error ↓ | $0.108 \pm 0.055$ | $0.113 \pm 0.06$ |
| | Root Mean Square Error ↓ | $0.154 \pm 0.078$ | $0.153 \pm 0.081$ |
| Triplet-based | Accuracy ↑ | $0.543 \pm 0.032$ | $0.642 \pm 0.065$ |
| | False Negative ↓ | $0.306 \pm 0.102$ | $0.181 \pm 0.038$ |
| | False Positive ↓ | $0.151 \pm 0.081$ | $0.177 \pm 0.029$ |
| | Matthews Correlation ↑ | $0.091 \pm 0.063$ | $0.284 \pm 0.13$ |
| | Mean Absolute Error ↓ | $0.204 \pm 0.109$ | $0.206 \pm 0.104$ |
| | Root Mean Square Error ↓ | $0.269 \pm 0.144$ | $0.271 \pm 0.138$ |

Table 4: To determine the optimal number of layers required for the best estimation of the embedding function, a comparison was conducted between two networks containing 5 and 14 layers, respectively.

