# OpenReview forum: "Bridging Causality, Individual Fairness, and Adversarial Robustness in the Absence of Structural Causal Model"
_TMLR — Accepted by TMLR_

### Review · Reviewer_P5Zm · 2024-10-27

**Summary Of Contributions:**

This paper presents a novel distance metric that combines the ideas of individual fairness and adversarially-robust learning and embeds both in a structural causal model. Briefly, this metric fulfills the requirements that for a feature space consisting of sensitive attributes $s:={s_1, s_2, … , s_k} \in S^{k}$ and non-sensitive attributes $r:={r_1, r_2, … , r_m} \in S^{m}$,

1.  $d((s, r) = d((s’, r)))$ for all $s, s^{‘} \in S, r \in R$
2. $\forall$  $s \in S, r \in R,  \exists \epsilon, \delta$ s.t.  $\forall \Delta \leq \delta, d(s, CF(r + \Delta)) \leq \epsilon$.

Where CF stands for the counterfactual "do" operartion.

In the paper, the authors motivate the need for a causal structure,  and describe how to compute this metric, by introducing the semi-latent spaces (in which the causal parents of sensitive attributes are removed). The authors then propose using deep learning to estimate the distance metric from labelled data, and then using this metric to perform fair learning.

**Audience:**

Yes

**Claims And Evidence:**

No

**Requested Changes:**

The below proposed adjustments must be addressed for securing this reviewer’s recommendation to acceptance.

When appropriate (most of the time), cite references using the Latex command \citep, which correctly encloses the reference in parentheses.

Definition 4.2. Why place the restriction that the metric be 0 only for twin pairs? Also, note that the metric definition does not place any restrictions in the case $d(s, r), (CF(s + \Delta), r)$. This seems undesirable.

Definition 4.3: “indigenous” -> “endogenous”. Also, “replacing the causal effects of parents of sensitive attributes and replacing their exogenous variables with endogenous ones” is unclear - possibly should be the reverse?   In the definition of $\varphi_i$, what is $F_i$? Also, please demonstrate why this mapping is actually a bijection. Also a typo: “There map bijective map”. Also, “the semi-latent space is simpler due to the independence of its components” - the components are not independent, since changing a sensitive attribute can have causal effects on the non-sensitive attributes. Also, in the same paragraph, what is the “intervention assumption”?

Section 5, 2nd line - what is meant by “data changes still fall the same category for the model”?

In the paragraph between Def. 5.1 and Prop. 5.2 - Note that two balls are not ‘isomorphic’ - two spaces with metrics may be isomorphic.

Section 6 - please be explicit (perhaps with an example) on how $\varphi$ is constructed.

The paragraph after Proposition 6.2 is not clear, but also seems circular - if we have a way to data with distances (i.e., a causal structural map), why is it necessary to use NNs or another method to approximate the metric?

The last paragraph on page 7: This leads to three main insights: the dimensionality of the embedding space, the guarantee of independence from coordinates in this space, and understandings about $d_{\chi}$ - what is meant by “the guarantee of independence”, and “the understandings”, and how does it follow from the rest of the paper?

Top of page 8: the activation functions $\sigma$ do not map from $\mathcal{R}^{d-1}$ to $\mathcal{R}^{d}$.

Proposition 6.3 - This proposition doesn’t seem relevant to the paper, and I don’t agree with the interpretation. First, note that the dimensions of the weight matrices are implicitly included in the norm vectors (more dimensions -> higher norms). Second, the Rademacher complexity deals only with the expressivity of the functions, and has no implication on learning dynamics.
Please explain the first formula in section 7 in words (e.g., what is $h_{\psi})? What is $\mathcal{P}_{\mathcal{D}}$ - earlier, $\mathcal{D}$ was defined as a set of observations - what does it mean for variables to be distributed according to a set of observations?

Likewise, please explain the formulas on the bottom of page 8/top of page 9 (as a side suggestion, it would have been helpful to number the equations). For example: What is $\dot{y}$? Does the star symbol stand for multiplication? What are $\mu_{1}$, $\mu_{3}$, and $\mu_{3}$?

First paragraph of section 8 - please define Siamese metric learning. Also please justify why it’s relevant to try three different types of data annotation, and propose a scenario in which such labeled data would be available, but the actual causal structure would not be known. If there is no such scenario (i.e., the causal structure is known), then what is the utility of estimating this distance with a neural network?

Second paragraph of section 8 - please explain what is meant by “designing the embedding network. What is meant by “half the input size for an unknown network?  Why is that a reasonable baseline? It seems far too large.

First paragraph of page 10 - “Although Prop 6.2 asserts that deep learning can converge with various layer sizes…” - it seems that the authors meant Prop. 6.3, but as detailed above, that proposition also does not assert this.

Please explain the experiments and results in Table 1.

When discussing Figure 3, please address the fact that in many cases CAPIFY performs much better than ESCAPIFY, especially on the Adult dataset.

Please clearly indicate how the adversarial robustness is measured, and what the results are.

In the Appendix proof of Proposition 5.2, please explain the statement that $CF(v, do(S:=s’)) = \ddot{v}_{s’}$. This seems incorrect, as the do-intervention can also impact other variables in the causal graph.

**Strengths And Weaknesses:**

Strengths:

I think that the paper tackles an interesting idea, in that there are some similarities in how we think about fairness versus adversarial robustness, and the idea of computing similarities in the context of a structural causal model very much strengthens the fairness framework. The background section is generally well-written.

Weaknesses:

Unfortunately the paper contains many weaknesses that renders it unsuitable for publication in its current form. The details of these are also outlined in the requested changes section; however, they are also summarized here.

* The paper contains many errors, making it very difficult to read.
* Sections of the paper use unclear language (e.g., this reviewer could not always understand what was meant by “embedding”)
* The motivation for the definition of the metric is not clear. Specifically, note that the metric definition does not place any restrictions in the case $d\big((s, r), (CF(s + \Delta), r)\big)$
* The proofs are not always clear
* The motivation for using a neural network for estimating the causal distance is unclear. Note that to learn this distance, it is required to have labelled data, for which a causal graph is required. If a causal graph is available, then why can the metric not be computed directly?
* The experimental section is very hard to understand, both in terms of what experiments are actually performed, and why. The adversarial robustness of the classifiers doesn’t seem to be evaluated

The errors, in particular, make the paper very difficult to follow and evaluate. Addressing these issues will make it far easier to evaluate the paper’s contribution.

---

> ### Author Response · Authors · 2024-11-15
>
> We thank the reviewer for their detailed feedback and constructive suggestions.
>
> 1. **Citation Formatting:**
>    We will ensure proper citation formatting using `\citep` for consistency.
>
> 2. **Definition 4.2:**
>    The restriction for twin pairs ensures fairness under counterfactual scenarios. With a sensitive attribute and causal structure, we expect the metric to behave consistently across data points and their counterfactual instances. For unrestricted cases, the metric must remain well-behaved and continuous:
>    $$\lim_{\Delta \to 0} d((s, r), CF((s, r), \Delta)) = 0.$$
>
> 3. **Definition 4.3 and Related Issues:**
>    The phrase "replacing the causal effects of parents of sensitive attributes and replacing their exogenous variables with endogenous ones" means removing the causal effects of sensitive attributes' parents and replacing their exogenous noise with endogenous variables in the new SCM. This preserves sensitive attribute variation.
>
>    In this definition, $F = g^{-1}$, where $g$ is the reduced-form mapping of the SCM. We assume $g$ is invertible, as required for identifiability and counterfactual identifiability. Bijective generation mechanisms (BGMs), including additive noise models, satisfy these conditions.
>
>    In the semi-latent space, non-sensitive components correspond to their exogenous counterparts and are independent by assumption. Arbitrary interventions on sensitive attributes imply independence in the semi-latent space.
>
> 4. **Section 5, 2nd Line:**
>    Revised text:
>    > "An adversarial perturbation ball defines a region in the input space where variations in data do not alter the model's predicted category."
>
> 5. **"Isomorphic" Expression:**
>    The perturbation ball combines regions, each isomorphic to a unit ball. However, the inclusion of sensitive attributes results in a disconnected structure, making it non-isomorphic to a single unit ball.
>
> 6. **Feedforward Neural Networks (Section 6):**
>    Classical feedforward neural networks are used, with logistic regression serving as a simple example.
>
> 7. **Proposition 6.2:**
>    Even if counterfactuals cannot be derived solely from observational data, fair metrics can be reconstructed using distances between observations. This avoids the need to fit an SCM directly, enabling practical applications with deep learning methods.
>
> 8. **Embedding Space (Page 7):**
>    The causal fair metric framework implies an embedding space with defined dimensions, independent coordinates, and a dissimilarity function. These properties guide the design of specialized neural networks for efficient learning.
>
> 9. **Top of Page 8:**
>    The expression $\sigma: \mathbb{R}^{d_{i-1}} \to \mathbb{R}^{d_{i-1}}$ refers to the index $i-1$, not a dimensional reduction.
>
> 10. **Proposition 6.3:**
>    Proposition 6.3 guarantees that deep learning-based metric estimation works effectively. While matrix norms embed dimensions theoretically, the sample-based guarantees are dimension-free. Here, $h_\psi$ is a parametric model (e.g., feedforward neural networks), and $\mathcal{P}_{\mathcal{D}}$ represents the empirical distribution.
>
> 11. **Counterfactuals:**
>    $\ddot{v}$ is counterfactual of data points $v$.  In the formula of bottom of page 8, $\mu_1$, $\mu_2$, and $\mu_3$ are regularizer coeffienets.
>
> 12. **Siamese Metric Learning:**
>    Siamese networks use a neural network $\phi$ to map inputs $v_1$ and $v_2$ into an embedding space, minimizing $d(\phi(v_1), \phi(v_2))$ for similar points and maximizing it for dissimilar ones. Contrastive loss is used:
>    $$\mathcal{L} = y \cdot d^2 + (1-y) \cdot \max(0, m - d)^2,$$
>    where $d = \|\phi(v_1) - \phi(v_2)\|$, $y$ indicates similarity, and $m$ is the margin.
>
> 13. **Method and Comparison:**
>    Our method estimates causal fair metrics without requiring a causal structure. By leveraging partial information (e.g., non-sensitive attributes and embedding independence), it outperforms traditional metric learning.
>
> 14. **Baseline Comparisons:**
>    Baselines assume an embedding space with smaller dimensionality than the feature space. We used half-space configurations, consistent with prior work.
>
> 15. **Table 1 Results:**
>    Table 1 shows our framework effectively estimates causal fair metrics, achieving high confidence across datasets, perturbation radii, and tagged data scenarios.
>
> 16. **CAPIFY vs. ESCAPIFY:**
>    CAPIFY performs better because it assumes knowledge of the causal metric, whereas ESCAPIFY first learns the metric before fitting the model.
>
> 17. **Adversarial Robustness:**
>    Robustness is measured by the "unfair area":
>    $$\mathbb{P}(\{x : \exists x' \text{ such that } d(x, x') < \Delta \text{ and } h(x) \neq h(x') \}).$$
>
> 18. **Equation $CF(v, do(S := s')) = \ddot{v}_{s'}$:**
>    Derived directly from the definition of twins under hard interventions, as explained in the background section.
>
> [1] A. Nasr-Esfahany et al., *Counterfactual Identifiability of Bijective Causal Models.*

---

> > ### Comment · Reviewer_P5Zm · 2024-11-18
> > **Thank you for the response - additional clarification requested.**
> >
> > Thank you for addressing some of my concerns. If it is possible, I would encourage the authors to upload an updated draft of the manuscript, since many of the concerns are issues of readability of the manuscript itself.
> >
> > My main remaining concern is how it is possible to obtain the training data for the distance metric estimation without assuming a causal structure in order to create distance labels. Could the authors please clarify how they do the distance labelling (both here and in the manuscript).
> >
> > Pointwise responses (using the authors' numbering) below.
> >
> > 2. Thank you for the explanation and I am sorry for misreading the definition of 'twin pairs' in the initial review. The definition makes sense.
> > 3. Please make these changes in the manuscript.
> > 6 and 7. I am not sure which of my points this is in response to. In particular, I am still not sure how distances between observations can be obtained (for labelling point distances and therefore learning the distance metric) without having a causal map in place. I assume that the distance would be as in Definition 4.2, please correct me (and clarify in the manuscript) if I am wrong on this point.
> > 9. Please clarify this in the manuscript.
> > 10. I still do not agree with the authors' interpretation of Prop. 6.3, for the reasons given in the original review. First, because I would argue that the implicit dimensional dependence in the formula impliles that the claim of dimension-independence is incorrect, and second, because the Rademacher complexity deals only with model expressivity and not function learnability. If the authors wish to provide additional results to address these points, they are encouraged to do so.
> > 11. Note that I asked about \dot(y) and not \ddot(v). Also, please put these explanations into the manuscript.
> > 12. Please put this explanation into the manuscript.
> > 14. (a) Please cite the prior work that also uses the half-size comparisons. (b) Regardless of the existence of this work, this seems like an odd choice since it is generally quite large, so the reader is left wondering if a smaller choice would have been more effective.
> > 15. Please add a better explanation of Table 1 into the manuscript.
> > 16. Note that Figure 3 is done on real-world data, implying that even for CAPIFY, the causal structure is presumed, not actually known, so using it as the 'golden' standard is somewhat flawed. However, my comment was simply that these gaps are not mentioned in the discussion in the paragraph in the top of Page 11, and I believe they should be.
> > 17. Please add this explanation to the manuscript, as right now there is nothing in the main body linking the adversarial robustness with the unfair area.

---

> > > ### Author Response · Authors · 2024-11-25
> > >
> > > Thank you for your feedback. We have implemented the improvements and provided clearer explanations as suggested in the revised version of the paper.
> > >
> > > Regarding your second point, the main contribution of our work is emphasizing that in robust learning, it is crucial to utilize a causal fairness metric derived from the causal structure. However, finding the exact causal structure is not always necessary, and in many cases, it is not feasible. Let us illustrate this with a simple example.
> > >
> > > Consider $M$, a linear structural causal model (SCM) with Gaussian noise. In this case:
> > > $$
> > > V = MV + U \implies U = (I - M)V
> > > $$
> > >
> > > From this, the causal fairness metric takes the structure:
> > > $$
> > > d(v_1, v_2) = \\| (I - M)v_1 - (I - M)v_2 \\|_p
> > > $$
> > >
> > > To learn this distance, we do not need to exactly estimate the causal structure. In the case of a linear SCM with Gaussian noise, the SCM itself is not identifiable. Instead, it suffices to estimate $(I - M)$ up to some rotation. For example, for any rotation matrix $O$, we still have:
> > > $$
> > > d(v_1, v_2) = \\| O(I - M)v_1 - O(I - M)v_2 \\|_p
> > > $$
> > >
> > > This demonstrates that finding the exact causal structure is not required to calculate the causal fairness metric. Instead, less information—specifically, $(I - M)$ up to a rotation—is sufficient.
> > >
> > > This highlights the advantage of our proposed method: we can bypass finding the exact causal structure by learning the fair metric directly from distance-labeled data. This makes our approach more practical and applicable in scenarios where determining the exact causal structure is challenging or infeasible.
> > >
> > >
> > > Regarding Point 5, we mentioned that other researchers, such as Kozdoba, Mark, and Shie Mannor, have used similar methods to demonstrate the convergence of metric learning. Specifically:
> > >
> > > - Kozdoba, Mark, and Shie Mannor. *"Dimension Free Generalization Bounds for Non-Linear Metric Learning."*
> > > - Kozdoba, Mark, and Shie Mannor. *"Two Regimes of Generalization for Non-Linear Metric Learning."*
> > >
> > > These works employ related approaches to establish generalization bounds and convergence for non-linear metric learning methods. By referencing these studies, we align our method with established literature and demonstrate its validity in the context of convergence analysis.

---

> > > > ### Comment · Reviewer_P5Zm · 2024-11-26
> > > >
> > > > 1. can you please propose a realistic scenario where the exact causal structure is not available, but the distance-labeled data is? My concern is that this data is impossible to obtain this data without using the causal structure. Conversely, if the causal structure is known, what is the benefit of estimating it with a neural network instead of using it directly?
> > > >
> > > > 2. Could you please explain the link between the results in these papers and convergence (of SGD or another algorithm)? I only see generalization results, which don't seem relevant to your work. If I am wrong about this, could you please explain the link?

---

> > > > > ### Author Response · Authors · 2024-11-29
> > > > >
> > > > > 1. A simple and realistic example of our method is based on the linear structural causal model, commonly known as the Structural Equation Model (SEM). If all exogenous variables in SEM follow a Gaussian distribution, it becomes impossible to uniquely identify the causal graph purely from observational data. However, through metric learning, even without prior knowledge of the causal structure, we can estimate a causal fair metric. SEM has widespread applications in fields such as Psychology, Sociology, Education, Econometrics, and Healthcare Management. Conversely, if the causal structure is known, our paper (section 6) demonstrates how to construct a causal fair metric directly, eliminating the need to estimate it using deep learning techniques. This highlights the flexibility of our approach in both known and unknown causal scenarios.
> > > > >
> > > > > 2. As discussed in our paper, our method provides general guarantees comparable to other deep learning approaches. However, establishing exact finite-sample guarantees for convergence requires further research and is beyond the scope of this work. The primary goal of our paper is to propose a framework that demonstrates how, in the presence of a causal structure, deep metric learning can simplify the process of deriving a causal fair metric. This serves as a foundation for future work to build upon, including deriving explicit theoretical guarantees.

---

### Review · Reviewer_fCeB · 2024-10-31

**Summary Of Contributions:**

The authors propose a causal fair metric to integrate fairness, robustness, and causality in AI, focusing on individual fairness and resilience to adversarial perturbations. This metric leverages causal structures involving sensitive attributes and creates protected causal perturbations without relying on structural causal models (SCMs), using metric learning instead for adaptability in real-world applications. Through experiments on synthetic and real datasets, their approach, ECAPIFY, demonstrates improvements in fairness and robustness within classifiers while preserving accuracy. Key contributions include (1) the causal fair metric, (2) protected causal perturbation, and (3) a practical SCM-independent metric learning framework.

**Audience:**

Yes

**Broader Impact Concerns:**

Nothing to add to the existing one.

**Claims And Evidence:**

Yes

**Requested Changes:**

1. Experimental Evaluation

- Results in Table 1 compare different learning scenarios; however, to fully support the claim that accuracy improves when embedding space dimensions and metric information are combined, the results should include a measure of variation to validate this improvement.

2. Improve Readability

- To enhance readability, remove duplicate citations (e.g., "Dwork et al. Dwork et al. (2012)" in Section 1).
- Proofread the manuscript to ensure consistent notation (e.g., Example 4.1 uses M and M' but also M_1 and M_2).
- Relocate the Broader Impact Statement to follow the Discussion and Future Work section for better flow.

**Strengths And Weaknesses:**

## Strengths

- The paper addresses the topic of fairness, robustness, and causality integration, which is a growing focus in the AI research community.

- The proposed causal fair metric and associated methods are thoroughly explained and well-supported by theoretical justifications and experimental evaluation.

- The paper is well-written and well-organized and discusses the method's limitations.


## Weaknesses

- Results reported in Table 1 are incomplete and require additional details (see Requested Changes - 1).

- The authors do not provide the source code.

---

> ### Author Response · Authors · 2024-11-15
>
> Thank you for your valuable feedback. We appreciate your suggestions for minor edits and will incorporate them in the final version to improve readability and clarity. Below, we provide detailed responses to your comments.
>
> Regarding Table 1, it is intended to provide an overview of our results. Due to space constraints, we included only the average of the measures. However, to illustrate the variation, we included error bars in Figures 2, 3, and 4 in the appendix. Additionally, we presented the detailed variations of the measures in Tables 2–4 in the appendix for further clarity.
>
> We also mentioned in the paper that the code is available on GitHub. We also attached the codes as supplemntary.

---

### Review · Reviewer_YuZs · 2024-11-18

**Summary Of Contributions:**

The paper proposes a framework for jointly learning a model that exhibits three important trustworthy AI properties: individual fairness, adversarial robustness, and causality. The proposed framework introduces a causal fair metric that encodes the causal structure and protected causal perturbation. This metric ensures that the similarity between individuals is both causally informed and sensitive to protected attributes. The authors then use metric learning to train classifiers that are both causally fair and robust. Experiments on real-world and synthetic datasets demonstrated the effectiveness of the method.

**Audience:**

Yes

**Broader Impact Concerns:**

The authors provided a Broader Impact Statement

**Claims And Evidence:**

Yes

**Requested Changes:**

- Citations in the paper must be improved. In some parts of the paper, the authors’ names are duplicated in the citations. For example, in the introduction, first paragraph, line 5, the reference name Dwork et al. is duplicated. The author should use only bib \cite and \citep properly without handwriting the author's name.

- The notations in the paper should be consistent to ease the reading. For example, in Example 4.1, the authors declare two structural causal models, $M$ and $M’$, and later state $M_1$ models the variables as independent, $M_2$ specifies a linear causal relationship. The authors should replace $M_1$ by $M$ and $M_2$ by $M’$.
In the same example, the authors should clarify the values of the gender distribution (e,g., {M, F}) and the corresponding numerical values. It was not clear to me how counterfactual change in gender in $M’$ yield $CF(v, F) = (F, 0, -1)$. How does the variable Income become -1 in under $M’$? And why is the $d(v, v_F)=3$ and not 4? Providing these clarifications can ease the comprehension of the example.
- In example 6.1, "_The counterfactuals for v with respect to $M_A$ and $M_A$ are (0, 0, 0) and (0, 0, N ), respectively_" it should be $M_A$ and $M_B$

- The authors should thoroughly define the fairness metrics used in the experiments, e.g., `UnfairArea`;

- Discuss how the learned robust model affects adversarial robustness across demographic subgroups.

- Since there is a lack of real-world datasets for metric learning causal structures, can the authors briefly discuss the importance of having such datasets in the real world and how they can be collected in practice?

- Having the Broader Impact Statement at the beginning of the Appendix or before the references would be better.

**Strengths And Weaknesses:**

Strengths:
- The paper is generally well-written and well-organized.

- The causal fair metric is intuitive and well-motivated with examples.

- The idea of learning the metric in causal latent space without directly using the structural causal models is novel and intuitive.

- The experiments are thorough and well-conducted.

- The paper clearly states the limitations of the proposed method and provides a detailed Broader Impact Statement.


Weaknesses:

- Some parts of the paper must be revised for clarity. Please see the requested changes below.

- The experiment did not clearly show how the proposed method improved individual fairness. See the requested changes for details.

- Definition 4.2 shows that the causal fair metric still relies on structural causal models, even when defined using the Semi-latent Space. How does the proposed method completely alleviate the need for structural causal models?

- Since the proposed methods ensure boat individual fairness and robustness, does it also alleviate the disparate robustness across groups, or does it exacerbate it [1]?


[1] Xu, Han, et al. "To be robust or to be fair: Towards fairness in adversarial training." International conference on machine learning. PMLR, 2021.**

---

> ### Author Response · Authors · 2024-11-18
>
> Thank you for your thoughtful and constructive feedback. We are honored that our work has captured your attention and have revised our manuscript to address your suggestions for improved clarity and rigor. Below, we outline the specific changes and provide detailed explanations for the issues raised:
>
> 1. **Citation Issue**: The problem with citations was related to the use of the `\cite` command. We have replaced all instances with `\citep` to resolve this issue.
>
> 2. **Notation Clarification**: The terms $M_1$ and $M_2$ have been corrected to $M$ and $M'$ in the revised version, where $F = 0$ and $M = 1$.
>
> To compute the counterfactual in a Structural Causal Model (SCM), we follow the **three-step procedure**:
>
> - **Abduction**: The noise variables are inferred as:
>   - $U_G = 1$
>   - $U_E = 0$
>   - $U_I = -1$
>
> - **Action**: Apply the intervention $G = 0$.
>
> - **Prediction**: Using the intervened SCM:
>   - $E = G + U_E = 0 + 0 = 0$
>   - $I = G + 2E + U_I = 0 + 2(0) - 1 = -1$
>
> Thus, the counterfactual point is $(G, E, I) = (0, 0, -1)$, and the distance $d((1, 1, 2), (0, 0, -1)) = |1 - 0| + |2 - (-1)| = 4$.
>
>
> 3. **Robustness Measure**: Robustness is evaluated by the "unfair area," defined as:
>    $$
>    \mathbb{P}({v : \exists v' \text{ such that } d(v, v') < \Delta \text{ and } h(v) \neq h(v') })
>    $$
>    Based on this measure, we also define the "counterfactual unfair area" and the "unrobust area."
>
> 4. **ECAPIFY Method**: In our ECAPIFY method, the model demonstrates robustness concerning the causal fairness metric, which incorporates our fairness criteria. After identifying a robust model, we ensure individual fairness by evaluating counterfactual fairness across different levels of sensitive attributes for each instance. Robustness serves as a foundation for achieving individual fairness in this approach.
>
> 5. **Tagged Data Collection**: Collecting tagged data aligned with specific metrics is becoming increasingly prevalent. For example, in new Large Language Model (LLM) methodologies, tagged data plays a critical role in providing explicit feedback on which responses better meet user needs. This feedback enables the fine-tuning of models for improved performance. Tagged data, with annotations highlighting the quality or appropriateness of responses, helps train LLMs to differentiate between effective and less effective outputs. We anticipate that collecting such data will become standard practice in real-world applications in the near future.

---

### Decision · Action_Editor_VFip · 2025-01-16

**Recommendation:** Accept with minor revision

**Comment:**

The paper proposes a framework for jointly learning a model that exhibits three important trustworthy properties: individual fairness, adversarial robustness, and causality. The proposed framework introduces a causal fair metric that encodes the causal structure and protected causal perturbation. This metric ensures that the similarity between individuals is both causally informed and sensitive to protected attributes. The authors then employ metric learning to train classifiers that are both causally fair and robust.

The majority of the reviews are favorable toward acceptance, though one reviewer raised several significant concerns. Below, I discuss the major issues raised.


> Concern #1. ‘The following claim (in the abstract) needs to be amended or removed: "To enhance the practicality of our metric, we propose metric learning as a method for metric estimation and deployment in real-world problems in the absence of structural causal models." The reviewer commented that while the SCM is not used in the metric learning framework, it is required to create distance-labeled training data for metric learning. In light of this, this claim is, at a minimum, highly misleading as written.

I agree with this concern. However, I believe it comes more from a writing issue than an overstatement of the actual contribution. In Proposition 6.2, the authors formally prove that, without the ground-truth SCM, data-driven metric learning is infeasible. This proposition adequately highlights the framework’s inherent limitations. Additional evidence supporting this point is also presented in the paper.

At the same time, I agree that revisions are needed to address this gap clearly. I suggest the following revisions:

- In the abstract and introduction, clarify the gap, the impossibility proposition, and the proposed approximation.
- In Section 6, re-organize the discussion to emphasize these gaps explicitly.

> Q#2. In Section 6, the use of Proposition 6.3 is not justified or explained. In particular, the authors do not explain the relationship between Rademacher complexity and "discern[ing] embeddings regardless of dimension" and it is not at all clear what that relationship might be, or what exactly the authors mean by "discerning". Further, the claim that the Rademacher complexity is independent of dimension is not justified without additional evidence, as the norms in Equation 7 scale with dimension.

I agree with this concern. The scope of Proposition 6.3, derived from other papers, feels somewhat awkward in this context. As noted by the reviewer, this section requires revisions to better explain the relationships mentioned and provide the necessary justifications.


**Decision**

After reviewing the concerns raised by the reviewer and considering feedback from other reviewers, as well as my own reading, I believe these issues, while valid, do not constitute substantial over-claiming. Most concerns arise from wording or narrative ambiguities rather than fundamental errors in the methodology or contributions. Therefore, I recommend minor revisions, focusing on improving the clarity and organization of key sections.

**Audience:**

Yes

**Claims And Evidence:**

Partially, see the comments for details